# Better continental-scale streamflow predictions for Australia: LSTM as a land surface model post-processor and standalone hydrological model

Ashkan Shokri[1], James C. Bennett[1], David E. Robertson[1], Jean-Michel Perraud[2], Andrew J. Frost[3], Eric A. Lehmann[2]

[1]Commonwealth Scientific and Industrial Research Organisation (CSIRO), Clayton 3168, Australia
[2]Commonwealth Scientific and Industrial Research Organisation (CSIRO), Mountain View 2460, Australia
[3]Australian Bureau of Meteorology (BoM), Sydney 2000, Australia

*Correspondence to*: Ashkan Shokri (ash.shokri@csiro.au)

**Abstract.** Accurate large-scale hydrological predictions are essential for water resource planning. However, many land surface models encounter difficulties in capturing streamflow timing and magnitudes, particularly in large catchments and when calibrated across broad regions and multiple hydrological variables. In this study, two Long Short-Term Memory (LSTM)-based approaches are assessed to enhance streamflow predictions across Australia: (i) LSTM-C, a standalone rainfall–runoff LSTM that relies solely on precipitation and potential evapotranspiration as inputs, and (ii) LSTM-QC, a rainfall–runoff LSTM that incorporates runoff outputs from the Australian Water Resources Assessment–Landscape model (AWRA-L), which can also be interpreted as a post-processor for AWRA-L. These approaches are tested in 218 minimally impacted catchments from the CAMELS-AUS dataset under three cross-validation strategies—temporally out-of-sample, spatially out-of-sample, and spatiotemporal out-of-sample—to evaluate their robustness for historical reconstructions, predictions in ungauged basins, and a proxy for climate-projection scenarios. The results indicate that both LSTM-QC and LSTM-C consistently outperform AWRA-L runoff across nearly all catchments and exceed the predictive skill of a widely used conceptual model (GR4J) in most basins. Under a temporally out-of-sample framework, LSTM-QC demonstrates a performance advantage over LSTM-C by leveraging information embedded in AWRA-L, particularly when fine-tuned to local catchment observed data. This advantage is primarily attributed to the LSTM's ability to correct systematic biases in AWRA-L and enhance channel-routing signals. However, under spatial and spatiotemporal cross-validation LSTM-C performs comparably well, suggesting that a purely data-driven approach can generalize effectively to ungauged or future conditions without reliance on AWRA-L.

## 1 Introduction

Traditionally, land surface models have been used to simulate key hydrological variables such as runoff, soil moisture, and evaporation across very large regions – often at continental scale. The widespread spatial coverage of these model predictions often trades off against accuracy: many land surface models face challenges in accurately capturing observed streamflow dynamics, particularly in large catchments, where they often perform worse than simple calibrated conceptual streamflow

models. The Bureau of Meteorology's (*the Bureau*'s) AWRA-L (Australian Water Resources Assessment – Landscape; Shokri et al., 2018; Frost & Shokri, 2021; Sharples et al., 2024), for instance, provides gridded hydrological outputs across the continent, but can perform poorly in simulating streamflow in comparison with calibrated conceptual rainfall-runoff (CRR) models (Frost et al., 2021). AWRA-L underpins the Australian Bureau of Meteorology's Australian Water Outlook service
(https://awo.bom.gov.au) that provides historical simulations (from 1911 until yesterday), seasonal forecasts, and long-range projections under climate change scenarios for a wide range of applications (e.g. antecedent conditions and seasonal outlooks for flood/fire risk, long term water availability assessment). Improving its predictions is thus likely to have significant benefits for Australian water managers and citizens. It is worth noting that while many land surface models were originally developed to provide boundary conditions for Earth system models rather than for direct streamflow prediction, AWRA-L was
specifically designed for water balance estimation and runoff prediction across Australia, with an emphasis on hydrological applications rather than atmospheric coupling, and calibrated to streamflow observations.

There are two main causes for AWRA-L's underperformance in relation to CRRs. First, AWRA-L is calibrated to multiple streamflow gauges, remotely sensed soil moisture, Evapotranspiration (ET) and Terrestrial Water Storage from Gravity Recovery and Climate Experiment (GRACE), with calibration carried out jointly for all gauges to a single objective for the
entire continent (Frost et al., 2021). The focus of this approach is on overall water balance, rather than one single component of the water balance (e.g. streamflow) and heterogeneity of the landscape means that individual site performance is not targeted. This means that streamflow simulation performance at any given gauge trades off against 1) overall performance at gauges across Australia and 2) performance at simulating variables other than streamflow. Second, AWRA-L does not attempt to simulate channel routing processes (e.g. routing delay, transmission losses); streamflow is simulated at a given point by
accumulating gridded runoff within a catchment area (it is noted that routing and losses and interactions with dams for water accounting purposes is simulated by a separate model, so AWRA-L was not designed to incorporate these processes). This means that the timing of streamflow peaks and recessions can disagree with observations, particularly in larger catchments. In addition, the sometimes-imperfect representation of hydrological processes in AWRA-L (and any model) reduces confidence in its streamflow predictions.  AWRA-L also does not explicitly simulate lakes or large reservoirs. While the catchments used
in this study are not impacted by major reservoirs and were nominally selected to avoid the impact of farm dams (see Zhang et al, 2013), small farm dams are widespread across many agricultural regions of Australia. These can significantly alter runoff and storage patterns, particularly during dry years, by reducing downstream flows. Although farm dams are not directly represented in AWRA-L, their effects are likely partially captured through calibration where observational data are available and farm dams were present. Recent studies have highlighted the growing regional impact of farm dams on water availability
under climate change (Malerba et al., 2021; Peña-Arancibia et al., 2023).

AWRA-L's lack of channel routing is common to several gridded land surface models, and accordingly past studies have highlighted the importance of improving streamflow routing, and/or adding processing methods to incorporate this process in these models. Wu et al. (2014) developed a coupled land surface and routing model that leverages real-time satellite-based precipitation data for global flood estimation. Their findings underscore the need to augment land surface models modeling

approaches to improve flood prediction accuracy. Similarly, Li et al. (2013) introduced a physically based runoff-routing model, demonstrating its effectiveness in simulating hydrological processes within land surface and Earth system models. Yassin et al. (2019) presented methods for improved representation of reservoir operations within hydrological models, advocating for improved parameterization methods. Yamazaki et al., (2011) introduced the CaMa-Flood model, which enhances floodplain representation by incorporating subgrid-scale topographic parameters. Their work demonstrates the importance of considering floodplain inundation dynamics in river routing models. These findings collectively indicate that enhanced routing techniques are important to improve simulations of streamflow dynamics in land surface models across a range of catchment sizes and characteristics.

Apart from physically based approaches for representing routing, several methods have been developed applying machine learning to estimate streamflow. For instance, Nagesh Kumar et al. (2004) used a feedforward Artificial Neural Network to estimate monthly flow time series of a single river. Recent advances in deep learning, particularly in Long Short-Term Memory (LSTM) networks, offer promising alternatives to traditional hydrological models. LSTMs are designed to model sequential data, making them well-suited for hydrological systems. When trained on observed data LSTMs have shown significant potential for streamflow prediction by directly learning from data rather than relying on predefined physical processes (Kratzert et al., 2018, 2019; Nearing et al., 2021). LSTMs have demonstrated significant potential both as standalone hydrological models and as post-processors to land surface models, where they improve streamflow routing and prediction. LSTMs offer at least two key advantages over conceptual rainfall-runoff or routing models. The first is their ability to accept novel predictors, allowing the easy incorporation of static and dynamic predictors without a fixed perspective on how they contribute to the representation of the hydrological process, potentially providing additional information to improve predictions. The second is their ability to 'learn' hydrological theory, when trained on a sufficiently large cohort of catchments (Nearing et al., 2021; Kratzert et al., 2024). A third advantage is that they are not constrained by physical laws such as mass balance, which allows them to implicitly correct biases in the input data. In contrast, in land surface models, uncertainty in the inputs typically propagates directly to the outputs.

LSTM models have been shown to be effective as post-processors of streamflow predictions from land surface models and other hydrological models. We will refer to these as *hybrid* approaches following Slater et al., (2023). Frame et al. (2021) demonstrated that LSTMs, when used as a post-processing technique on the U.S. National Water Model, markedly improved predictions. Also in the U.S., Konapala et al. (2020) showed that hybrid models that combine physically based model outputs with LSTMs can significantly enhance streamflow simulation capabilities across a range of catchments. Yu et al. (2024) implemented a hybrid approach, applied over the Great Lakes region of North America, called the Spatially Recursive (SR) model, which integrates a lumped LSTM network with a physics-based hydrological routing simulation. They demonstrated enhanced streamflow prediction capabilities. This approach outperformed standalone lumped LSTM models, especially for large basins and ungauged basins, by considering spatial heterogeneity at finer resolutions. Interestingly, LSTM models have also been successfully applied in cascade configurations, where multiple LSTM models are stacked in sequential layers, with the outputs of one layer serving as inputs to the next. This approach is particularly useful for medium-range streamflow

forecasts, as it allows the model to first predict intermediate variables, such as precipitation, which are then used to refine the final streamflow predictionClick or tap here to enter text.Click or tap here to enter text..

While hybrid approaches improve streamflow predictions from land surface models, the combination of land surface model and LSTM may not outperform a standalone LSTM. For example, Frame et al. (2021) found that using a standalone LSTM produced more accurate predictions in ungauged basins than a hybrid land surface model-LSTM. It thus remains an open question as to which land surface model-LSTM hybrids are worthwhile, and which would be better replaced with LSTM-only models. The value of hybrid approaches may depend on the application context. For example, hybrid models may be particularly valuable for climate change scenario analysis, where maintaining physical consistency with land surface model outputs is important. Conversely, standalone LSTM models may be more advantageous for applications such as prediction in ungauged basins, where maximizing data-driven performance is the priority. In addition, in cases where LSTMs improve predictions from land surface model, the source of the improvements may be interpreted as the information content provided by the land surface model. Land surface models often exhibit two main deficiencies: routing errors and systematic biases in specific catchments. Routing errors primarily affect the timing and shape of the hydrograph, while systematic biases reflect consistent over- or underestimation of flow magnitude, regardless of timing. By comparing hybrid models trained with short input sequences (i.e. one time step) to those trained with longer sequences, we can distinguish the type of information AWRA-L contributes. Short sequence lengths limit the LSTM's ability to correct routing errors, meaning improvements in this case are more likely due to information related to bias correction.

The major aim of this study is to assess the performance of streamflow predictions from an AWRA-LSTM hybrid, which has never been previously assessed. We use both static attributes (e.g. fixed catchment characteristics such as catchment area) and dynamical predictors (streamflow predictions from AWRA-L, precipitation, potential evaporation) to construct the AWRA-LSTM hybrid. Different approaches have been previously adopted to establish hybrid land surface model and LSTMs (e.g. Frame et al., 2021; Lima et al., 2024; Tang et al., 2023). We therefore investigate how best to implement the AWRA-LSTM hybrid by refining the method we use to apply the LSTM, including the choice of dynamic and static predictors. Once the model is developed, we are able to diagnose the relative contributions of bias-correction and routing improvements. A secondary aim of this study is to establish the performance of LSTMs both as a post-processor for AWRA-L and as a standalone hydrological model in Australia. While we expect previous findings from other studies to be replicated – e.g. that standalone LSTMs will generally outperform conceptual rainfall-runoff models for predictions in ungauged basins (Kratzert et al., 2019) – replicating these findings in Australia is an important precursor to the adoption of LSTMs for a broad range of uses here.

To rigorously test our findings, we test performance in 218 catchments from the CAMELS-Aus dataset (Fowler et al., 2021). We evaluate the performance of the model using spatial and temporal out-of-sample cross-validation to assess its generalization capability. The cross-validation experiments test LSTM post-processed AWRA-L predictions for three applications:

- Predictions in ungauged basins. AWRA-L is regularly applied for continental scale water accounting and water forecasting, including in ungauged basins.

- Predictions in gauged basins for periods outside the gauge record. A key application of AWRA-L is to assess long-term trends in the historical hydrological function of Australian catchments (e.g. Ho et al., 2023; Wasko & Nathan, 2019).

- Predictions in ungauged basins for periods outside of gauged records. AWRA-L is used to generate long-range climate projections, including in ungauged basins, and this cross-validation strategy serves as a proxy for testing climate projection capabilities.

In each experiment, AWRA-LSTM predictions are compared with the unprocessed accumulated runoff from AWRA-L and a high-performing conceptual rainfall-runoff model in GR4J which has been extensively applied and evaluated in the Australian context. Coron et al., (2012) provide a comprehensive evaluation of GR4J performance across 216 Australian catchments under diverse climate conditions. Hapuarachchi et al., (2022) describe the use of GR4J as part of the operational ensemble streamflow forecasting system for Australia. Zheng et al., (2024)further demonstrate the application of GR4J in projecting future streamflow under various climate change scenarios for Australia.

This study aims to contribute to these ongoing questions by evaluating the performance of an AWRA-LSTM hybrid, assessing its strengths and limitations as both a diagnostic tool and a predictive model, particularly for Australian hydrological contexts. Additionally, by exploring these dynamics, we aim to inform the broader application of hybrid and standalone models, guiding future hydrological modeling efforts.

## 2 Methods

### 2.1 Data

#### 2.1.1 AWRA-L predictions

AWRA-L runs on a daily time step on a 0.05° grid, with national historical outputs available from 1911 onward. AWRA-L v7 (most recent iteration of AWRA-L ) was calibrated using data from 295 catchments over the period 1981–2011, employing a objective function that incorporates weighting of the following observations in each catchment GRACE Terrestrial Water Storage (TWS: 50%), streamflow (35%), satellite-based soil moisture (7.5%), satellite-based evapotranspiration (ET: 2.5%), and satellite-based vegetation fraction (5% ), which are then combined further over all catchments (Frost and Shokri, 2021). AWRA-L version 7 was extensively validated against a range of observational datasets (Frost et al., 2021. AWRA-L generates a number of variables that can potentially serve as predictors (e.g. evaporation, soil moisture, runoff, deep drainage). In this study, we focused only on runoff ($denoted\ Qtot$). In AWRA-L, $Qtot$ is derived from surface flow, baseflow, and interflow. The discharge from these sources is routed (at a pixel scale) through a conceptual surface water store, $Sr$. The primary function of this store is to replicate the partially delayed drainage of storm flows, which is typically observed in all but the smallest and fastest-responding catchments. As noted in the introduction, however, the model lacks channel routing (with independent grid

cells with no lateral flow), creating challenges when calculating the streamflow at the outlets of large basins (i.e. those with a time of concentration greater than one day).

Streamflow at catchment outlets/gauges was calculated by summing $Qtot$ from all grid cells within the catchment, weighted by the proportion of each grid cell within the catchment. However, this approach does not account for in-stream routing, stream losses, overbank flow, or storages. The time series of the accumulated runoff for each catchment was used for both benchmarking (to compare as a baseline methodology) and as a dynamic predictor for the LSTM model.

### 2.1.2 CAMELS-AUS

The CAMELS-AUS dataset (Fowler et al., 2021, 2024) consists of streamflow, meteorological variables, and various
catchment attributes (222 Australian catchments in the version 1 and 561 in the version 2) for catchments that have been minimally impacted by human activities. We used the following data from the CAMELS-AUS:

**Streamflow**: This serves as the predictand and is used to evaluate model performance over the period 1975 - 2014 in CAMELS-AUS version 1 and 1975 - 2022 in CAMELS-AUS version 2.

**Rainfall**: CAMELS-AUS includes two time series of catchment-averaged rainfall. We used the "awap_rain" product, which
is taken from the Bureau's Australian Gridded Climate Data (AGCD). AGCD is produced at a spatial resolution of 0.05° (~5 km) by interpolating data from its extensive network of meteorological stations.

**Potential Evaporation (PE):** Among several evaporation products available, we selected "et_morton_wet_silo" from CAMELS-AUS. This product estimates potential evaporation under wet conditions using the Morton wet environment method, which accounts for factors such as temperature, humidity, wind speed, and solar radiation. The data is provided by the
Queensland Government's SILO database, offering an upper limit of evaporation potential.

**Static Attributes**: Static attributes are assumed to be constant over time and provide essential catchment characteristics. These attributes include mean annual precipitation, mean annual PE, aridity, and other climatic and geomorphological features such as average slope and catchment area. These static attributes help contextualize the dynamic data and improve the model's ability to generalize across different catchments.

**Catchment Boundaries:** CAMELS-AUS provides catchment delineations. The catchment boundaries were used to subset the AWRA-L gridded dataset to calculate total discharge at gauges as daily timeseries.

To illustrate the diversity of catchments used, Figure 1 shows their spatial distribution across Australia overlaid with the Köppen–Geiger climate classification (Stern et al., 2000).

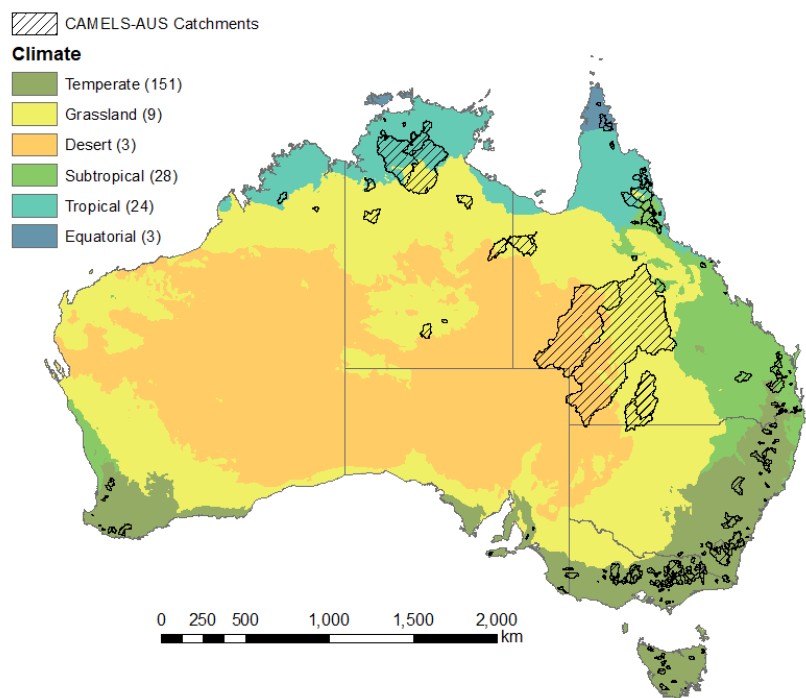

**Figure 1 Spatial distribution of CAMELS-AUS v1 catchments used in this study, overlaid on the Köppen–Geiger climate classification. Numbers in parentheses indicate the number of catchments in each climate class.**

## 2.2 LSTM model

In this study the LSTM model architecture (Hochreiter and Schmidhuber, 1997) was based on the work of Kratzert et al. (2019). The LSTM structure comprises four key components - an input gate, forget gate, cell state, and output gate -which collectively manage information flow and maintain the network's long-term memory (Figure 2).

The input sequence at each timestep includes one or more dynamic predictors. At the end of the sequence, the hidden state of the LSTM network is used to predict a single streamflow value as the target via a dense layer with one hidden layer of size 10. For all experiments, an LSTM hidden state size of 256 was used, along with smooth-joint Nash-Sutcliffe Efficiency (NSE) (Kratzert et al., 2019) loss function, and a sequence length of 365 days. All dynamic predictors were standardized with the mean and standard deviation of the calibration data across all catchments.

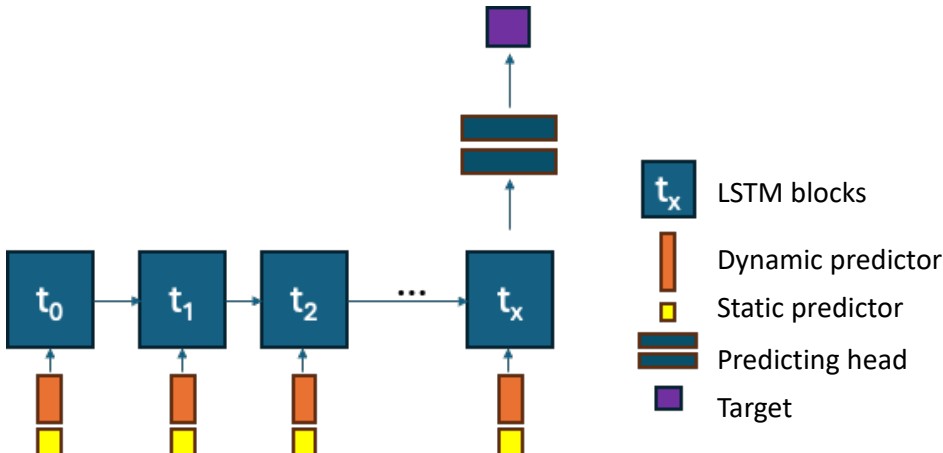

**Figure 2. Diagram of the LSTM model structure used in this study.**

**2.3 Predictors and target variables**

The LSTM models use spatially averaged $Q_{tot}$ from AWRA-L, and rainfall and ET from CAMELS-AUS as dynamic predictors as well as 12 static predictors, which represent the characteristics of each basin (Table 1). Additionally, the sine and cosine of the day of the year (*doy*), calculated as $\sin\left(\frac{2\pi doy}{365}\right)$ and $\cos\left(\frac{2\pi doy}{365}\right)$, are included as dynamically varying predictors to capture seasonality. The target variable is gauged daily streamflow from the CAMELS-AUS dataset, which is normalized by catchment area and expressed in millimetres.

**Table 1 Static and quasi static features for LSTM Model**

| Category | Predictor | Description |
|---|---|---|
| **Climatic and Precipitation Characteristics (static)** | p_mean | Mean Annual Precipitation |
| | pet_mean | Mean Annual Potential Evapotranspiration |
| | Aridity | Aridity (Mean Annual PET/Mean Annual Precipitation) |
| | p_seasonality | Precipitation Seasonality |
| | high_prec_freq | Frequency of High-Precipitation Days ($\geq$ 5 times mean annual) |
| | high_prec_dur | Average Duration of High Precipitation Events |
| **Catchment and Geomorphological Characteristics (quasi-static)** | catchment_area | Catchment Area |
| | mean_slope_pct | Catchment Mean Slope |
| | prop_forested | Proportion of Catchment Occupied by Forest |
| | Upsdist | Maximum Flow Path Length Upstream |
| | Strdensity | Ratio of Total Length of Streams to Catchment Area |
| | Strahler | Strahler Stream Order at Gauging Station |

## 2.4 Evaluation

### 2.4.1 Cross-validation approaches: TooS, SooS and TSooS

To evaluate the performance and generalizability of the LSTM models, three cross-validation techniques were employed: buffered Temporal out of Sample (TooS), Spatial out of Sample (SooS), and Temporal-Spatial out of Sample (TSooS). Each technique was designed to evaluate the model under different scenarios of data availability and variability. In addition, we reserved a separate hold-out test period (2014–2022), drawn from CAMELS-AUS version 2, which lies outside both the AWRA-L and LSTM calibration periods and was not used in training or validation.

**TooS – buffered temporally out-of-sample cross-validation:** This approach divides the entire dataset into k temporal folds (in this case, k = 4). Where for each fold 10 years data out of 40 years overall were designated as the validation set, and the remaining periods were used to train the model (Table 3). A trailing buffer of five years was applied after the validation period and subsequent training period to prevent data leakage and ensure temporal independence. This process was repeated for each fold, with each period serving as the validation set, while the model was trained on the other periods. After running the validation across all the folds, the results from each fold were combined to create a complete set of simulations produced in the validation mode.

**Table 2 Buffered temporal out of sample cross-validation folds**

| Fold | Training Period | | Validation Period |
|---|---|---|---|
| Fold 0 | 1/1/1990-31/12/2014 | | 1/1/1975-31/12/1984 |
| Fold 1 | 1/1/1975-31/12/1984 1/1/2000-31/12/2014 | and | 1/1/1985-31/12/1994 |
| Fold 2 | 1/1/1975-31/12/1994 1/1/2010-31/12/2014 | and | 1/1/1995-31/12/2004 |
| Fold 3 | 1/1/1975-31/12/2004 | | 1/1/2005-31/12/2014 |

**SooS – spatially out-of-sample cross-validation:** In this approach, the dataset is divided into four spatial groups, with each group containing a unique set of catchments. The model was trained on data from three of these groups and validated on the remaining group. This process was repeated for each fold, allowing each group of catchments to serve as a validation set once. Care was taken to ensure that nested catchments (i.e., catchments that share upstream-downstream relationships within the same river system) are grouped together. This prevents cases where hydrologically similar regions appear in both training and validation sets, which could lead to data leakage (a situation where the model learns patterns from similar catchments in the training set, leading to overly optimistic validation performance). To mitigate this, each nested group was assigned exclusively to either training or validation, ensuring independence between the two.

After all folds were used as the validation set, the validation results from each fold were combined to produce a complete assessment of model performance across all catchments. This method tests the model's ability to generalize to new, unseen spatial regions, making it suitable for evaluating its adaptability to areas with limited or no training data (Figure 3).

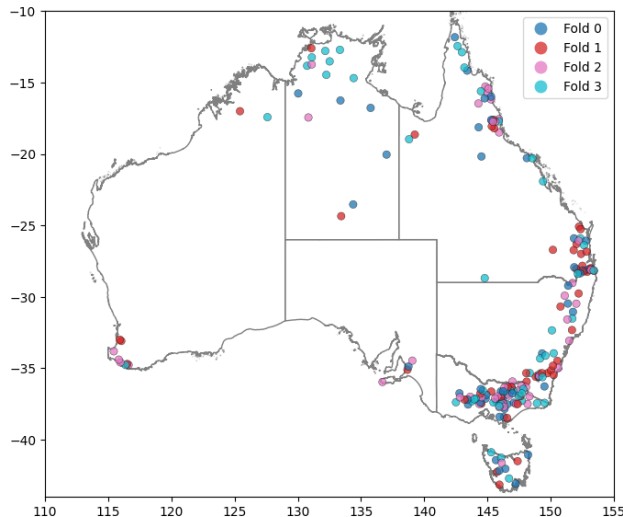

**Figure 3. Spatial distribution of catchments used in SooS cross-validation**

**TSooS – temporally and spatially -out-of-sample cross-validation:** The TSooS approach combines both temporal and spatial cross-validation to provide the most stringent assessment of model performance. A 2x2 fold-splitting technique was used, where folds 0 and 1 and folds 2 and 3 from the TooS and SooS experiments were merged, effectively dividing the dataset into four quadrants with both spatial and temporal splits. This setup allows the model to be trained in one quadrant and validated in a non-adjacent quadrant, ensuring that the validation data remain distinct from the training data in both time and space. This 245 method is strict because it requires the model to work well with new times and locations, making it a strong way to test the model's ability to handle different and realistic hydrological situations.

### 2.4.2 Evaluation Metrics

We evaluated model performance primarily using the Nash–Sutcliffe Efficiency (NSE; Nash and Sutcliffe, 1970), which measures how well predicted streamflow matches observations relative to the mean of the observed series. NSE is defined as

$$NSE = 1 - \frac{\sum_t(y_t-\widehat{y}_t)^2}{\sum_t(y_t-\bar{y})^2},$$ (1)

where $y_t$ and $\widehat{y}_t$ are the observed and predicted discharged at time $t$, and $\bar{y}$ is the mean observed discharge. NSE provides an overall measure of performance across all time steps but is more sensitive to high flows. To provide complementary information on other parts of the hydrograph, we also report NSE calculated in square-root space and the absolute bias in appendices. For each catchment, the best-performing model was defined as the one achieving the highest NSE, without 255 applying a buffer.

## 2.5 Experimental Design

### 2.5.1 Model Design

We examine how different combinations of dynamic predictors influence the performance of our LSTM models in predicting streamflow. Two experiments were conducted:

- The first experiment included both the AWRA-L output (i.e., gridded runoff from surface and subsurface processes at a 5 km × 5 km resolution across Australia) and additional climate variables, specifically rainfall and evaporation, as predictors in the LSTM. Although these climate variables (or at least very similar estimates of precipitation and potential evapotranspiration) are used within the AWRA-L model, we hypothesized that incorporating them directly might retain valuable information. Specifically, transformations of these climate variables may not be adequately captured by the AWRA conceptualization, leading to conditional errors. By incorporating these variables, we aimed to correct such errors and enhance the LSTM's predictive skill. This model will be referred to as LSTM-QC.

- The second experiment completely bypassed AWRA-L, using only climate variables (rainfall and potential evapotranspiration) as predictors. In this setup, the LSTM essentially acts as a rainfall-runoff model, eliminating any dependence on AWRA-L. This model will be referred to as LSTM-C.

In addition to these dynamic predictors, both models also used catchment attributes to provide spatial context. Two categories of attributes were considered:

- Static predictors: geomorphological characteristics that are independent of climate and can be assumed to remain stable over time if no major developments occur in the catchment area. These include features such as catchment area, mean slope, proportion of forest cover, stream density, Strahler order, and maximum upstream distance.

- Quasi-static predictors: climatic characteristics, such as mean annual precipitation and potential evapotranspiration, which may change due to long-term climate variability or climate change. For temporal cross-validation (TooS), it is important to consider the period over which these variables are calculated. By default, the climatic attributes provided with CAMELS-AUS are calculated for the entire available record. However, for proper TooS cross-validation, it is necessary to exclude the validation period and recalculate these climatic variables for each fold. This prevents information leakage and ensures that predictors reflect only the calibration data. Recalculation is particularly important when training the model in a wet or dry period and testing its generalizability to an opposite condition.

To avoid confusion, we therefore refer to geomorphological attributes as static predictors, since they do not vary across time windows, and to climatic attributes as quasi-static predictors, since they must be recalculated for different time periods.

The full set of predictors used in each model configuration is summarized in Table 3, with the detailed definitions of static and quasi-static predictors provided in Table 1.

**Table 3 Predictors used in LSTM-C and LSTM-QC models**

| Model | Dynamic predictors | Static and quasi-static predictors |
| --- | --- | --- |

| | | |
|---|---|---|
| LSTM-C | • Rainfall,<br>• Potential evapotranspiration | • Climatic attributes<br>• Geomorphological attributes<br>(see Table 1) |
| LSTM-QC | • Rainfall<br>• Potential evapotranspiration,<br>• AWRA-L runoff | • Climatic attributes<br>• Geomorphological attributes<br>(see Table 1) |

### 2.5.2 Training approach

The initial experiment involved calibrating a continental LSTM model using data from all catchments simultaneously. This calibration enabled the model to learn general patterns and relationships across diverse geographical and climatic conditions.

The training data were prepared using a sliding window approach with a sequence length of k=365 days. For each catchment n, the dynamic predictors at time t are represented by the feature vector $\mathbf{x}_{t,n} \in \mathbb{R}^d$ (rainfall, potential evaporation, AWRA-L runoff, and seasonality terms), and the static attributes are represented by $\mathbf{a}_n \in \mathbb{R}^p$ (climatic and geomorphological characteristics). For each sample, the input sequence is the matrix $\mathbf{X}_{t,n} = [\mathbf{x}_{t-k+1,n}, \ldots, \mathbf{x}_{t,n}] \in \mathbb{R}^{k \times d}$, which together with $\mathbf{a}_n$ is used to predict observed discharge of day $t$, $y_{t,n}$. This produced overlapping samples that were shuffled across all catchments

to form a diverse training set.

The LSTM parameters $\boldsymbol{\theta}$ were optimized to minimize the smooth-joint Nash–Sutcliffe Efficiency loss function (Kratzert et al., 2019), $\ell(\hat{y}_{t,n}, y_{t,n})$, where predictions are given by $\hat{y}_{t,n} = M_{\boldsymbol{\theta}}([\mathbf{X}_{t-k+1,n}, \ldots, \mathbf{X}_{t,n}])$.

To evaluate whether the general knowledge gained from the continentally calibrated model could be further enhanced at the individual catchment level in TooS experiments, an additional fine-tuning process was implemented. After developing the

300 continental-scale model, before implementing it for validation, further training of all model parameters is conducted using only the data from the calibration period of each catchment. This step allows the model to better capture localized patterns by adjusting its parameters to reflect the unique characteristics of individual catchments. In the SooS and TSooS cross-validation experiment, fine-tuning for individual catchments would not be realistic in a true out-of-sample scenario, as no target catchment data would be available for adjustment.

To recognize the uncertainty in the training process, each calibration and validation was repeated 10 times, and the median of the 10 simulations was used to calculate performance metrics.

### 2.5.3 Decomposing the effect of bias-correction and routing

Any improvement achieved through post-processing the AWRA-L outputs can originate from two primary sources: correcting systematic model errors (bias-correction) and addressing temporal misalignments caused by the absence of channel routing in

the AWRA-L model. To investigate the relative contributions of these two sources, two versions of the LSTM post-processors with different predictor sequence lengths were designed.

All experiments described thus far use a sequence length of 365 days, allowing the model to capture temporal dependencies and account for flow routing—an effect that occurs over several days as water moves through river systems. This configuration is expected to correct both systematic biases and routing errors. Additionally, a series of experiments using shorter sequence lengths (i.e., 1, 2, 3, 4, 5, 10, 30, and 60 days) was conducted to analyze the sensitivity of the model predictive performance to this variable. When a single day is used, the model primarily focuses on correcting immediate daily discrepancies in the outputs, addressing bias without capturing temporal flow patterns.

By comparing the performance of these models, the sources of improvement can be decomposed. Significant sensitivity to sequence length would indicate that fixing temporal dependencies—and thereby correcting routing errors—contributes substantially to the model's enhanced accuracy. Conversely, minimal performance differences would suggest that most improvements are attributable to bias-correction alone.

## 2.6 GR4J

To test the performance of our AWRA-LSTM hybrid setup, we compare it to the GR4J conceptual rainfall-runoff model (Perrin et al. 2003). GR4J is a four-parameter model, developed through a rigorous process of parameter reduction to enable strong performance with automated calibration algorithms. It has been widely tested in Australia and abroad, often outperforming other conceptual rainfall-runoff models in automated calibration experiments (Coron et al. 2012). For this study, we optimize GR4J with shuffled complex evolution (Duan et al. 1993). To ease optimization and to enable parameters to be applied to different catchments under spatial cross-validation studies, we scale and transform GR4J parameters (see Appendix 0). In all cases, GR4J is initialized for 5 years before parameters are optimized.

For SooS and TSooS cross-validation experiments, we use a distance-weighted regional estimation (Regional) to produce GR4J parameters. For each recipient catchment we estimate a global parameter-set from $N$ donor catchments by maximising an inverse-distance weighted objective:

$$NSE_{global} = \sum_{n=1}^{N} w_n NSE_n \tag{1}$$

$$w_n = \frac{(1/d_n)^\alpha}{\sum_{i=1}^{N}(1/d_i)^\alpha} \quad \alpha \geq 1 \tag{2}$$

where $NSE_n$ is the Nash-Sutcliffe Efficiency for the $n$th donor catchment, $w_n$ is the weight applied to the $n$th donor catchment such that $\sum_{n=1}^{N} w_n = 1$, and $d_n$ is the Euclidean distance between the catchment centroid of the target catchment and the $n$th donor catchment. $\alpha$ controls the emphasis on nearby catchments: the higher the value, the more emphasis is put on more closer catchments. We choose $\alpha = 2$ for this study. We found that this regionalisation method tended to outperform a conventional 'nearest-neighbour' regionalisation in cross-validation experiments (not shown for brevity).

 **3 Results**

This section evaluates the performance of the LSTM models under different configurations and cross-validation strategies. It first examines model development, assessing the effect of fine-tuning, dynamic predictor selection, and the inclusion of static predictors. The next part benchmarks model performance across three applications: long-term historical simulations, predictions in ungauged basins, and a proxy for climate projections, while also analyzing spatial patterns in model performance.
The final part investigates systematic error correction and the influence of sequence length on model predictions.

**3.1 Post processor refinement**

**3.1.1 Effect of finetuning**

Figure 4 shows the performance of the LSTM-QC model according to NSE using the national LSTM-QC without fine tuning, then with local fine tuning. The left panel shows the NSE probability exceedance curve for 218 catchments, with the fine-
tuned model (blue) consistently outperforming the global model (red), especially at higher exceedance probabilities of NSE values across catchments. The right panel maps where each model performs best: blue points indicate gauges where fine-tuning outperformed, while red points show the opposite. The inset pie chart reveals that 95.4% of catchments benefited from fine-tuning. These results demonstrate the effectiveness of fine-tuning for better model adaptation to local conditions. It should be noted that in the four catchments where performance decreased after fine-tuning, two are ephemeral rivers with highly
intermittent flow regimes, and two have relatively short or limited records. These conditions make them particularly difficult to model, as extreme variability can lead to overfitting during fine-tuning, meaning the generalized global model may sometimes perform better. From here on, only fine-tuned results are presented for TooS cross-validation.

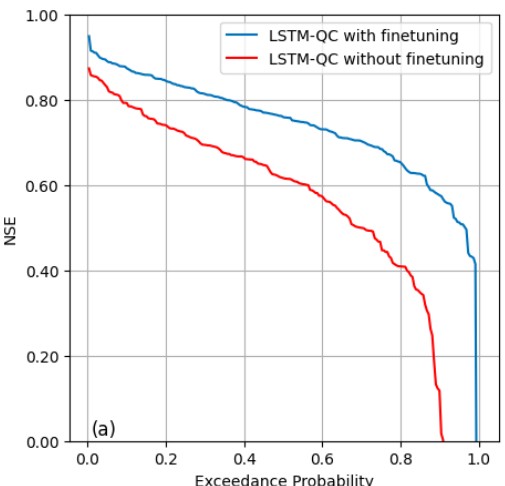 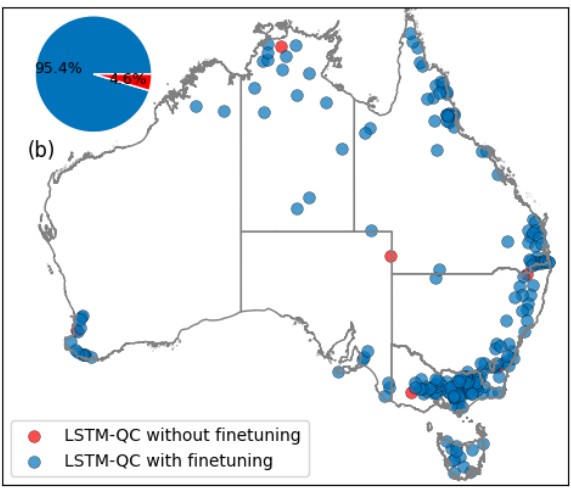

**Figure 4. NSE probability exceedance curve over all 218 catchments (left), comparing the performance of the pretrained and finetuned LSTM-QC model when using temporal out of sample (TooS) cross validation. Curves closer to 1 show better performance.**
**Right panel is a map of where each model performed best; blue (red) points show gauges where LSTM-QC with finetuning**

**outperformed (underperformed) LSTM-QC without finetuning; inset pie chart shows the proportion of catchments in which LSTM-QC with finetuning outperformed (underperformed) LSTM-QC in blue (red).**

### 3.1.2 Dynamic predictor selection

The LSTM was used in two forms: i) as a post-processor for AWRA-L, where we used AWRA-L $Q_{tot}$ along with climate data
(rainfall and ET) as predictors (LSTM-QC); and ii) as a rainfall-runoff model without dependency on AWRA-L, using only climate data as dynamic predictors (LSTM-C). Figure 5 illustrates the effect of selecting dynamic predictors on model performance. In each catchment, the best-performing model is defined as the one with the highest NSE value. To avoid reliance on marginal differences, the exceedance curves also show the proportion of catchments where performance gains exceed any given threshold, providing a clearer picture of whether improvements are both consistent and substantial. LSTM-QC performs
better in the TooS experiment for 66.1% of catchments (Figure 5-a, and d). However, in the SooS and TSooS experiments, adding AWRA-L $Q_{tot}$ as a predictor generally does not improve performance compared to the LSTM-C predictions (Figure 5b, c, e, and f).

AWRA-L is calibrated over the period 1970 to 2011. Therefore, when using AWRA-L's $Q_{tot}$ as a predictor in a TooS cross-validation, there is a potential risk of information leakage from the calibration phase into the cross-validation. To determine if
this leakage significantly contributes to the observed improvement of LSTM-QC over LSTM-C, the calibrated model for the third fold (calibrated for 1975 to 2004) was used to simulate flows outside the AWRA-L calibration period, from 2012 to 2022, using the CAMELS-AUS V2 dataset. The dashed lines in Figure 5-a confirms that LSTM-QC outperforms LSTM-C for 2012-2022 in the TooS experiment as the difference is greater than zero for approximately 60% of catchments, similar to the general results.

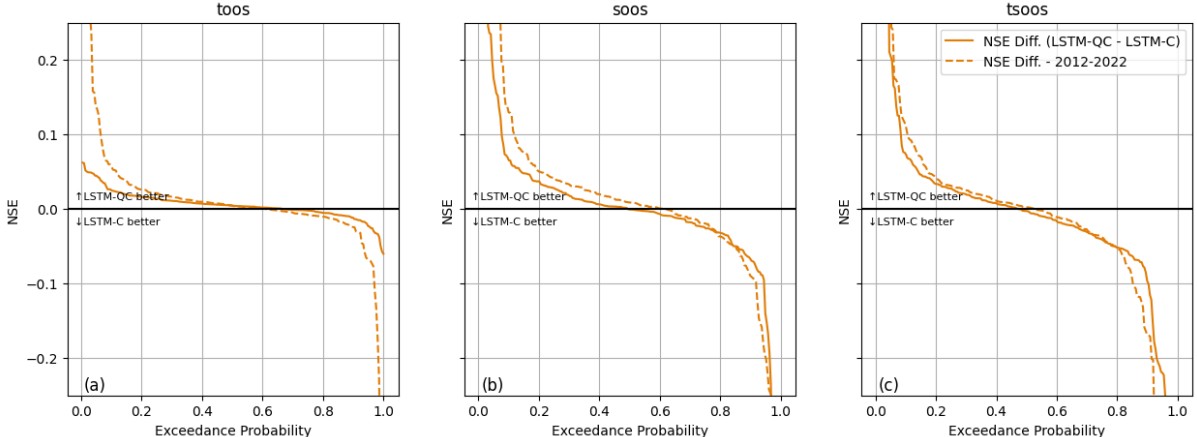

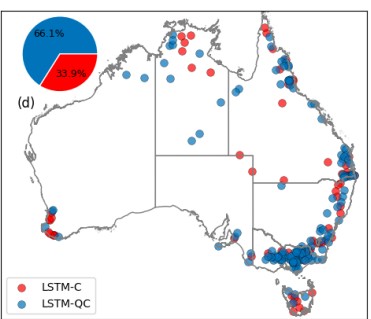 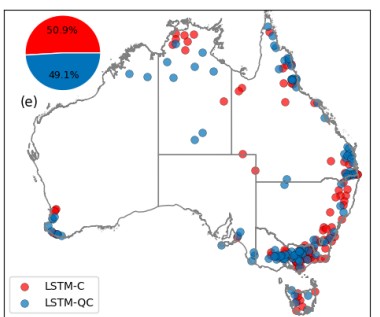 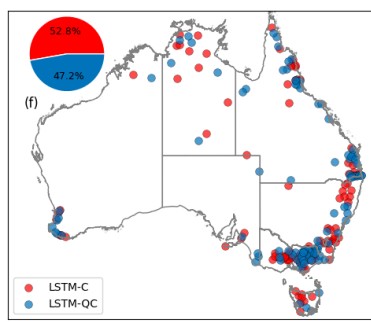

**Figure 5. Exceedance curve of NSE difference between LSTM-QC, which uses both AWRA-L $Q_{tot}$ and climate dynamic predictors, with LSTM-C, which uses only climate dynamic predictors, illustrating the information gain from including AWRA-L $Q_{tot}$. Top row shows exceedance probability curves of the NSE difference between LSTM-QC and LSTM-C under TooS (a) SooS (b) and TSooS (c) experiments. Bottom row shows spatial distribution of the best-performing model under each.**

### 3.1.3 Static predictors

Figure 6 compares the effect of cross-validation of static climatic variables (see section 2.5.2) on model performance when using TooS cross-validation, both without (Figure 6a) and with (Figure 6b) fine-tuning. Recalculating the quasi-static climatic variables for the calibration period at each fold slightly but consistently reduces the performance of predictions. Consequently, we cross-validate climate predictors for TooS and TSooS experiments in the remainder of the paper.

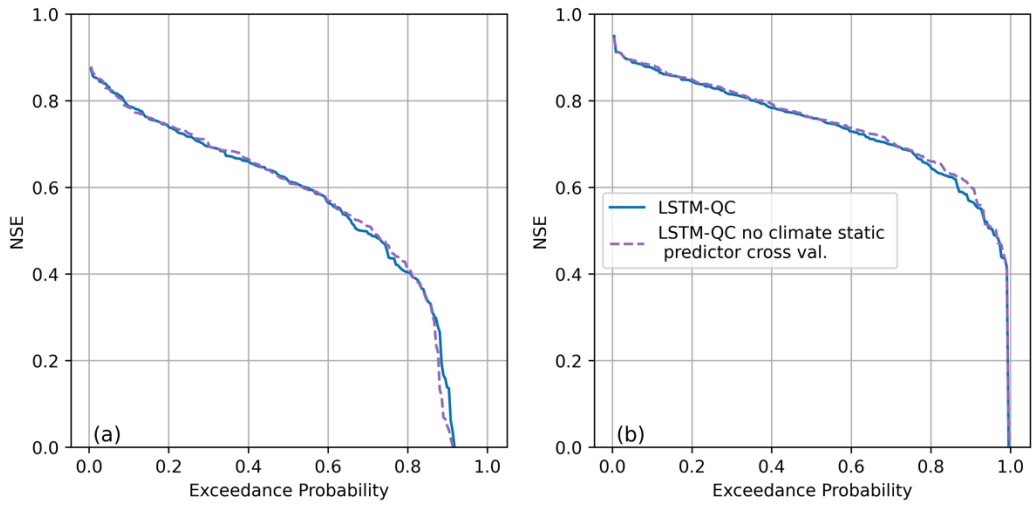

**Figure 6. Impact of cross-validation on model performance with (right) and without (left) fine-tuning of recalculated quasi-static climatic variables for each fold in TooS validation**

### 3.2 Applications

The LSTM model was evaluated across three distinct applications to assess its versatility and performance in different hydrological contexts. We benchmarked the LSTM-QC model against both the raw AWRA-L and GR4J simulations, focusing on: 1) long-term historical simulations in gauged catchments (TooS cross-validation), 2) predictions in ungauged basins (SooS

cross-validation), and 3) a proxy for climate projection capabilities (TSooS cross-validation). Figure 7 presents the benchmarking NSE results across these applications. The NSE in square root space and the absolute bias are presented in the Appendix.

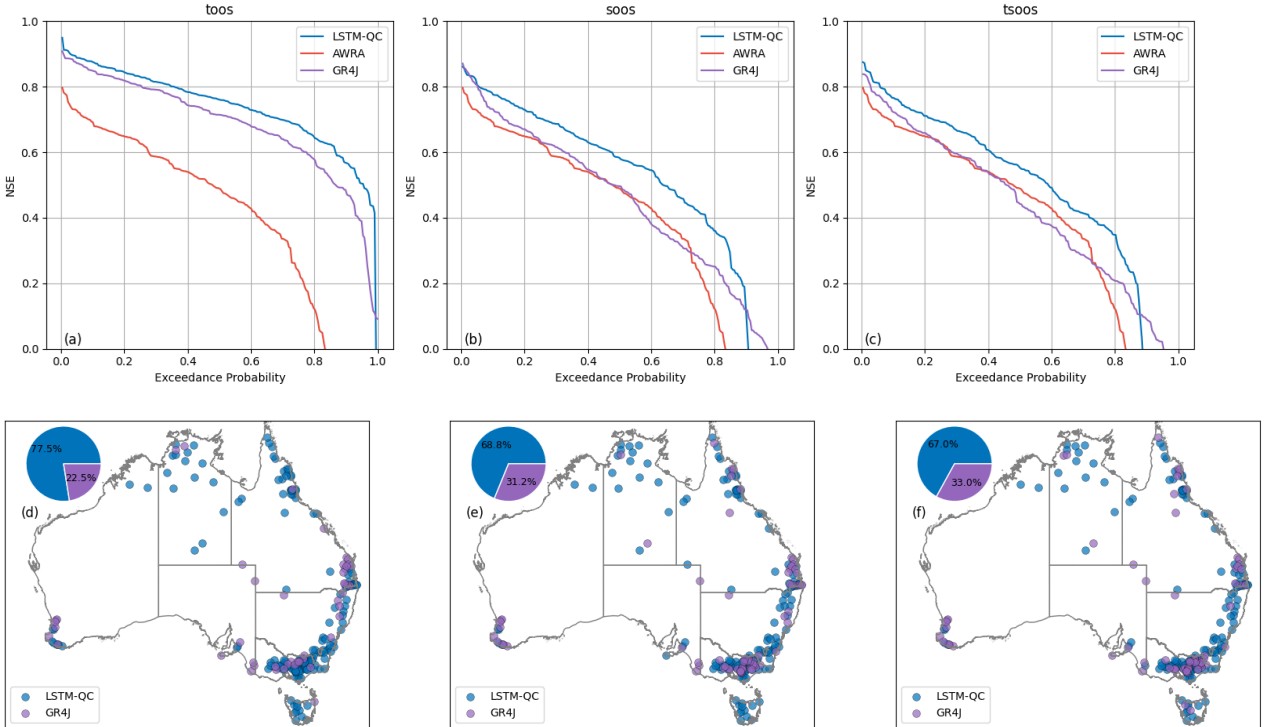

**Figure 7. Benchmarking LSTM-QC results against AWRA and GR4J. Top row is excedance curve of NSE across all catchments;**
**bottom row shows which model performs best for each catchment. The columns from left to right show TooS, SooS, and TSooS cross-validation experiments.**

### 3.2.1 Application 1 – long term historical simulation (TooS)

LSTM-QC with fine-tuning significantly outperformed GR4J in TooS experiments (Figure 7a, Figure 7d), achieving superior results in 77.5% of catchments. Both LSTM-QC and GR4J performed considerably better than the AWRA-L model. It should
be noted that AWRA-L employs a continental-scale calibration approach with a single set of parameters derived nationwide, while both GR4J and the fine-tuned LSTM-QC benefit from catchment-specific calibration. Thus, we do not expect AWRA-L to perform well in relation to the other models and included AWRA-L only as a reference.

### 3.2.2 Application 2 – predictions in ungauged basins (SooS)

LSTM-QC generally outperformed both GR4J and AWRA-L models in SooS experiments (Figure 7b, Figure 7e). Specifically, in 69% of catchments LSTM-QC performed better than GR4J. However, the regionally calibrated GR4J preforms better than LSTM-QC in catchments with poorer NSE. The regional calibration applied to GR4J is particularly adept at avoiding very poor performance, a notable advantage over both AWRA-L and LSTM-QC.

### 3.2.3 Application 3 – a proxy for climate projections (TSooS)

The TSooS results were largely consistent with those observed in the SooS approach (Figure 7c, Figure 7f): in 67% of catchments, LSTM-QC outperformed regionally calibrated GR4J.

This experiment is categorized as TSooS because validation data are independent in both space and time. Specifically, the model is trained on half of the catchments over half of the available period and tested on the other half of the catchments during the remaining period. This setup ensures that the validation set comprises entirely unseen catchments and time periods, providing a more stringent test of model generalization than TooS and SooS.

### 3.2.4 Spatial pattern

While the LSTM-QC model generally outperformed GR4J across all three applications, certain regions showed a clear advantage for the GR4J model. In areas such as Western Australia and the western parts of Victoria—characterized by unique hydrological behaviors (Grigg and Hughes, 2018; Saft et al., 2015)—the GR4J model demonstrated superior performance. Comparing the TooS cross-validation (which involves fine-tuning) and the other two (SooS and TSooS) shows that fine-tuning improves the performance of LSTM in these regions. These findings highlight the potential limitations of a highly generalized LSTM approach in regions with distinct hydrological dynamics.

### 3.3 Systematic Error Correction and Routing Impact of LSTM Performance as a Postprocessor

Figure 8 shows the performance of LSTM-C and LSTM-QC models for different LSTM predictor sequence lengths under a TooS cross-validation. At a sequence length of one, the performance of LSTM-QC for the median and upper band is similar to AWRA, but catchments with lower performance show improvement when LSTM-QC was used. This suggests that bias correction has little effect on the upper 50% of catchments, but for the lower tail of the distribution, LSTM improves AWRA through bias correction. The median and upper tail of the distribution improve after a sequence length of three, showing an improvement in performance metrics, which is mostly due to the channel routing processes and additional lag processes such as percolation, groundwater interactions, and human influences (e.g., farm dams). The performance of LSTM-QC improves considerably when the sequence length is increased from 1 to 365 days.

Conversely, LSTM-C performs poorly at very short sequence lengths. This is unsurprising: without the ability to attenuate climate forcings or catchment/channel routing processes, we do not expect LSTM-C to be able to simulate streamflow

efficiently. The different responses of LSTM-QC and LSTM-C to sequence length suggests that AWRA-L's built-in hydrological processes capture at least some of the long-term hydrological memory through its storage components. However, the LSTM-QC model's continued improvement with longer sequences indicates its ability to compensate for AWRA-L's lack of in-stream routing capabilities.

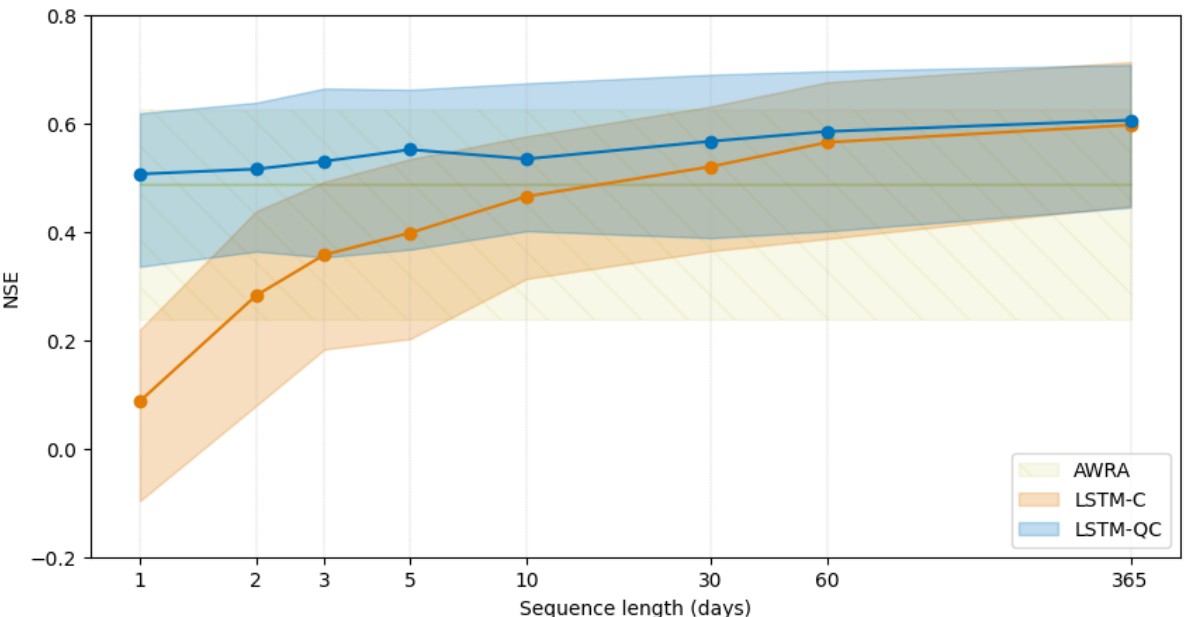

**Figure 8. Performance of LSTM-C and LSTM-QC across different predictor sequence lengths under a TooS cross-validation showing the information gain from including AWRA-L runoff.**

The comparison with LSTM-C at very short sequence lengths is not intended as a practical configuration for standalone LSTMs, which are known to require longer sequences to perform well. Instead, it serves as a baseline to illustrate the added value of AWRA-L runoff: even when the sequence length is too short for LSTM-C to capture catchment memory, LSTM-QC can leverage AWRA-L to resolve routing processes.

## 4 Discussion

The results of this study highlight the significant potential of LSTM networks in streamflow prediction as a rainfall-runoff model (LSTM-C) or as a rainfall-runoff model that incorporates additional information from AWRA-L land surface model (LSTM-QC), which can also be viewed as a post-processor for AWRA-L. The comparative analysis between LSTM and conceptual models, such as AWRA-L and GR4J, reveals the strengths and limitations of each approach, shedding light on the trade-offs between model complexity, and performance. Through a series of experiments, we have demonstrated how LSTM models can improve streamflow estimation across various regions and for different applications including prediction in gauged catchments, ungauged catchments and for projection studies.

LSTM models strongly outperform AWRA-L in streamflow prediction. Notably, predictive performance of LSTM-C models, which rely solely on climate data, surpass AWRA-L. However, incorporating AWRA-L outputs into the LSTM models (LSTM-QC) provides measurable information gain in temporal out of sample results by leveraging additional hydrological information provided by AWRA-L. This suggests that in gauged catchments, benefits can be gained from using AWRA-L as an input to LSTM models. However, there is a trade-off between complexity and gains in predictive performance, as LSTM-QC's slight advantage may not always justify the additional data requirements of an LSTM, especially when aiming for generalized, scalable models suitable for ungauged basins.

Kratzert et al. (2019) showed that using static catchment characteristics as predictors improves the LSTM results when trained over multiple catchments. In our study, two types of static predictors—geomorphological and climatic characteristics—were employed to specify catchment characteristics. Geomorphological features, being independent of climatic factors, are considered stable over time unless significant changes, such as land-use alterations or infrastructure development, occur within the catchment area. These features provide a reliable foundation for catchment characterization. Conversely, climate characteristics, such as mean precipitation and evapotranspiration, are subject to long-term changes due to climate variability and long-term climate change. As these predictors are more sensitive to temporal shifts, they require careful consideration when used in predictive models, particularly in the context of temporal cross-validation procedures. We observed slight but consistent improvements in performance when the static variables were calculated over the entire available period (including validation period), which is coming from information leakage from validation to calibration period, compared to when the calculation of these variables was cross-validated. Accordingly, we recommend that when a temporal split is involved in cross validation, i.e. TooS and TSooS cross-validation, it is necessary to recalculate the static climatic variables for each fold.

Similarly, care has to be taken when using land surface model outputs if they have been calibrated (as is the case with AWRA-L). Because they are computationally intensive, it is often infeasible to subject land surface models to multiple re-calibrations to carry out rigorous cross-validation. This was the case in our study. This raises the concern that when AWRA-L is used as a predictor, it could potentially transfer information from the validation period to the training period through its calibrated parameters. Since AWRA-L calibration relies on a single set of parameters across all catchments in Australia, any information leakage is likely to be minimal. To assess whether the observed improvements from incorporating AWRA-L were due to information leakage or if AWRA-L was genuinely enhancing model performance, a simulation for the 2011 to 2022 period (outside of AWRA-L's calibration range) was conducted. The results showed that the LSTM-QC model, which includes AWRA-L as a predictor, still slightly outperformed the LSTM-C model under TooS cross-validation. Accordingly, we recommend that out-of-temporal sample testing be conducted when using a land surface model as a predictor, if possible.

The comparison between fine-tuning and global calibration for the LSTM models using TooS cross-validation showed that fine-tuning significantly improved model performance. This localized fine-tuning allowed the model to better capture catchment-specific hydrological patterns, improving its predictive accuracy. These findings are consistent with previous studies, which have also highlighted the advantages of tailoring models to local conditions to enhance predictive performance (Frame et al., 2021; Kratzert et al., 2018). In contrast, global calibration, which uses a single model trained on all catchments

without further adjustment, showed lower performance, especially when applied to unseen catchments with distinct climatic and geomorphological characteristics. However, fine-tuning was not applied in SooS and TSooS cross-validation due to the lack of validation catchment data in the calibration phase, highlighting the advantages of fine-tuning when such data is available. It should be noted that regionalized fine-tuning using nearby catchments could be a viable alternative, although it was not implemented in this study.

The sensitivity of the LSTM-QC and LSTM-C to the length of predictors passed to the model was investigated, enabling a decomposition of the information provided by AWRA-L in terms of bias-related signals and routing-related signals. The results highlight distinct patterns of improvement achieved by the LSTM across different catchment types. In well-performing catchments, where the AWRA-L benchmark already demonstrates relatively high accuracy (NSE > 0.5), the primary benefit of the LSTM model is to correct timing errors in streamflow predictions. When an LSTM model is applied with a sequence length of just one day, its capacity to capture the temporal dynamics of streamflow routing is limited. Consequently, improvements in these cases are primarily attributed to systematic error correction rather than advancements in routing, confirming that the LSTM model cannot significantly surpass the AWRA-L benchmark in such catchments without explicitly addressing routing and timing. We also note that the influence of sequence length is partly modulated by catchment size: larger catchments tend to benefit more from the inclusion of AWRA-L runoff at very short sequence lengths, as its partial surface water storage provides additional memory. However, this advantage diminishes as longer sequences allow the LSTM itself to capture the relevant dependencies

Furthermore, the findings indicate that the LSTM using the AWRA-L predictor (LSTM-QC) in TooS cross validation outperforms climate-only predictors (LSTM-C) providing more accurate streamflow predictions due to its integration of detailed hydrological processes. The AWRA-L predictor implicitly includes the effects of multiple storage mechanisms, specifically three soil layers, groundwater and surface water storages, and therefore contributes to a deeper understanding of catchment water flow and retention. Consequently, the LSTM-QC model requires only a shorter backward-looking window since much of the necessary memory for slow routing processes is already embedded within AWRA-L's structure. However, a slight performance boost is observed by extending the sequence length beyond 5–10 days, particularly in LSTM-QC, suggesting that for some catchments enhanced slow routing processes are necessary.

The demonstrated superior performance (through TooS cross validation) of LSTM-QC in long-term historical simulation has significant implications for water resource management and planning. This capability is particularly valuable for water accounting studies, environmental flow assessments, and infrastructure planning in gauged catchments. The model's ability to outperform GR4J while maintaining consistent performance across multiple runs suggests that LSTM-QC is likely to produce more reliable assessments of long-term water balances. This robustness is especially crucial for applications such as reservoir operation optimization, where accurate long-term simulations of historical flow are essential for developing operational rules. The enhanced performance of the fine-tuned LSTM-QC also makes it suitable for retrospective analysis of extreme events and their impacts on water resources, providing water managers with a more reliable tool for understanding historical catchment

Behavior and improving future management strategies. However, there is a need to analyze in depth the predictions made of extreme events so as to be certain of the model's robustness and its applicability in various scenarios.

The LSTM outperformed GR4J under all cross-validation experiments for the majority of Australian catchments. This is a noteworthy outcome: GR4J is a widely used and high-performing rainfall-runoff model in Australian conditions. Perhaps the least surprising of these is the SooS performance, as LSTMs have been shown in a variety of studies to outperform conceptual models for predictions in ungauged basins (Frame et al., 2021; Kratzert et al., 2019). This capability has broad practical applications, particularly in remote areas and developing regions where gauge networks are sparse. The LSTM model's ability to outperform traditional GR4J simulations derived from regional calibration suggests its potential for improving water resource assessments in ungauged catchments, supporting applications such as small-scale hydropower development, irrigation planning, and flood risk assessment. The consistent performance across different catchment types indicates that the model more successfully captures the underlying hydrological processes and their spatial variations than existing alternatives, making it a valuable tool for regional water resource planning and management in data-scarce regions. It should be noted that in regions such as Western Australia and the western parts of Victoria —characterized by distinct hydrological behavior (Hughes et al., 2012; Petrone et al., 2010)—the GR4J model demonstrated superior performance. We attribute this primarily to the regionalization scheme employed in GR4J, which places greater weight on nearby catchments and therefore captures local hydrological signals that the LSTM, trained more globally, does not. While suboptimal LSTM training due to limited exposure to relevant catchment attributes may also contribute, we expect that training on larger or more diverse datasets could reduce this gap. We conclude that LSTMs should at least be considered for applications for which conceptual rainfall-runoff models are currently used in Australia.

The successful validation of LSTM using TSooS cross validation demonstrates its potential for supporting climate change adaptation strategies in water resource management. This capability is particularly valuable for infrastructure design and long-term water security planning. The maintained performance advantage in both spatial and temporal transferability indicates that LSTM could be effectively employed in climate impact assessments, supporting decision-making for adaptation measures such as reservoir design, environmental flow provisions, and urban water supply planning under various climate change scenarios. Moreover, this capability extends to regional-scale climate change vulnerability assessments, where understanding potential hydrological responses across multiple ungauged catchments is crucial for developing robust adaptation strategies.

While the results of this study highlight the advantages of LSTM models, it is important to acknowledge the challenges associated with their application. One key consideration is the computational and data overhead associated with training LSTM models on large datasets, especially for applications focusing on single catchments. In such cases, simpler models like GR4J may offer a more practical alternative without the need for extensive computational resources. Additionally, the application of LSTMs, as implemented in this study, focuses primarily on improving predictive skill rather than exploring hydrological hypotheses. Unlike conceptual models, which are designed to test causal relationships and provide insights into hydrological processes, LSTMs function as data-driven tools that excel in capturing patterns but are less suited for unravelling the sensitivity of runoff generation to specific predictors. Conceptual and land surface models also have the advantage of providing outputs

for a range of other hydrological variables (e.g., soil moisture, evapotranspiration, groundwater storage), in addition to streamflow. This highlights a trade-off between prediction accuracy and the ability to explore system dynamics, which must be considered when selecting models for specific purposes.

## 5 Conclusion

The potential of Long Short-Term Memory (LSTM) networks to enhance streamflow predictions in Australia was evaluated. The findings demonstrate that LSTM networks —whether functioning as standalone rainfall–runoff models (LSTM-C) or as rainfall–runoff models that incorporate additional information from AWRA-L (LSTM-QC), and thus can also be interpreted as post-processors to AWRA-L—consistently improve prediction accuracy across Australia relative to existing models. LSTM models outperformed traditional approaches, including AWRA-L and GR4J, particularly in applications involving ungauged basins, historical data analysis of gauged basins, and a proxy for climate projection scenarios.

This study highlights the applicability of LSTM-based hydrological models and post-processors in climate adaptation strategies, long-term water resource planning, infrastructure design, environmental flow provisions, and regional vulnerability assessments, especially in data-scarce or climatically dynamic regions. The results confirm that LSTM networks, when fine-tuned to specific catchments, effectively correct systematic biases and address routing deficiencies in AWRA-L, achieving superior predictive performance in gauged catchments. For TooS cross-validation, fine-tuning yielded notable improvements, particularly in catchments with less accurate AWRA-L predictions. Under SooS and TSooS cross-validation, which precluded individual fine-tuning, LSTM models benefited from the model's generalization capabilities derived from broader datasets.

Incorporating AWRA-L outputs into LSTM models (LSTM-QC) provided marginal gains over standalone LSTM-C models for TooS validation, with no obvious improvement in SooS and TSooS validation. This suggests that while AWRA-L contributes some hydrological insights, the additional complexity may not always justify its inclusion, at least in a catchment scale streamflow application as trialed here. The study underscores the importance of recalculating quasi-static climatic predictors, such as mean precipitation, during temporal cross-validation to avoid information leakage. Using static climate variables calculated over the calibration and validation periods together can compromise validation accuracy. Recalculation of these predictors for each fold ensured that the model's performance reflected its true predictive capabilities.

Integrating AWRA-L outputs with LSTM models provided additional hydrological insights by incorporating processes such as soil moisture storage and groundwater flow, which improved predictions for shorter memory windows. This integration was particularly effective in correcting systematic biases and routing errors, enhancing the representation of hydrological processes beyond what climate data alone could achieve.

Overall, the research establishes the utility of deep learning, particularly LSTM networks, in refining outputs from land surface models like AWRA-L. Future work should investigate incorporating dynamic predictors beyond runoff to further improve LSTM models' capacity to capture complex hydrological processes.

**Code and data availability**

The datasets used in this study are publicly available as follows: The AWRA-L dataset is available from the Australian Bureau of Meteorology (BoM). For access, visit the Australian Water Outlook website: https://awo.bom.gov.au/. SILO provides long-term climate datasets, including rainfall and potential evapotranspiration, from the Queensland Government. Access the SILO database at: https://www.longpaddock.qld.gov.au/silo/. The CAMELS-AUS dataset, including hydrometeorological timeseries and catchment attributes, is available through Earth System Science Data. The dataset can be accessed via: https://doi.org/10.5194/essd-13-3847-2021 (version 1) and https://doi.org/10.5194/essd-2024-263 (version 2). The code developed for this study is available upon request from the corresponding author due to licencing requirements.

**Author contributions**

AS, JCB, and DER designed the study. AS developed the models and performed the analyses. JMP provided software and technical support for model development. AJF and EAL provided data and contributed to interpretation. All authors contributed to the discussion of results and to the writing of the manuscript.

**Competing interests**

The authors declare that they have no conflict of interest.

**Acknowledgment**

This research was conducted on the traditional lands of the Boonwurrung people of the Kulin Nation. We pay our respects to their elders. We also acknowledge the traditional owners of the catchments used in this study. This research was supported by the Commonwealth Scientific and Industrial Research Organisation (CSIRO) AquaWatch Australia Mission and the CSIRO AI4Missions program. Additional support was received from the Murray-Darling Basin Sustainable Yields project. The authors also wish to express their appreciation to Dr. Nagur Cherukuru for his invaluable contributions to the management and coordination of this work. We acknowledge the use of artificial intelligence tools for partial proofreading of this manuscript.

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

**Appendices**

**GR4J parameter adjustments and transformations**

Our implementation of GR4J differs slightly from that described be Perrin et al. (2003). When inferring parameters for each catchment, we scale parameters as follows:

$$\tilde{x}_1 = x_1 \tag{A.1}$$

$$\tilde{x}_2 = 0.67 \times x_2 \tag{A.2}$$

$$\tilde{x}_3 = 2.21 \times x_3 \tag{A.3}$$

$$\tilde{x}_4 = x_4 \frac{\sqrt{C}}{250} \tag{A.4}$$

where $C$ is the catchment area. These scalings are based on our experience and on the advice of the developers of GR4J to maximize performance. To ease inference, we apply the following transformations to the parameters:

$$\log_{10}(\tilde{x}_1) \tag{A.5}$$

$$asinh(\tilde{x}_2) \tag{A.6}$$

$$\log_{10}(\tilde{x}_3) \tag{A.7}$$

$$\log_{10}(\tilde{x}_4) \tag{A.8}$$

For SooS experiments, when applying parameters from donor catchments to recipient catchments, we have to account for differences in catchment size between the donor and recipient catchments for $\tilde{x}_4$, as follows:

$$x_{4,d} = \tilde{x}_{4,d} \frac{250}{\sqrt{C_d}} \tag{A.9}$$

$$\tilde{x}_{4,r} = x_{4,d} \frac{\sqrt{C_r}}{250} \tag{A.10}$$

where $\tilde{x}_{4,d}$ is the $\tilde{x}_4$ parameter from the donor catchment and $C_d$ is the catchment area of the donor catchment, and $\tilde{x}_{4,r}$ is the converted $\tilde{x}_4$ parameter used in the recipient catchment and $C_r$ is the catchment area of the recipient catchment.

**Other performance metrics:**

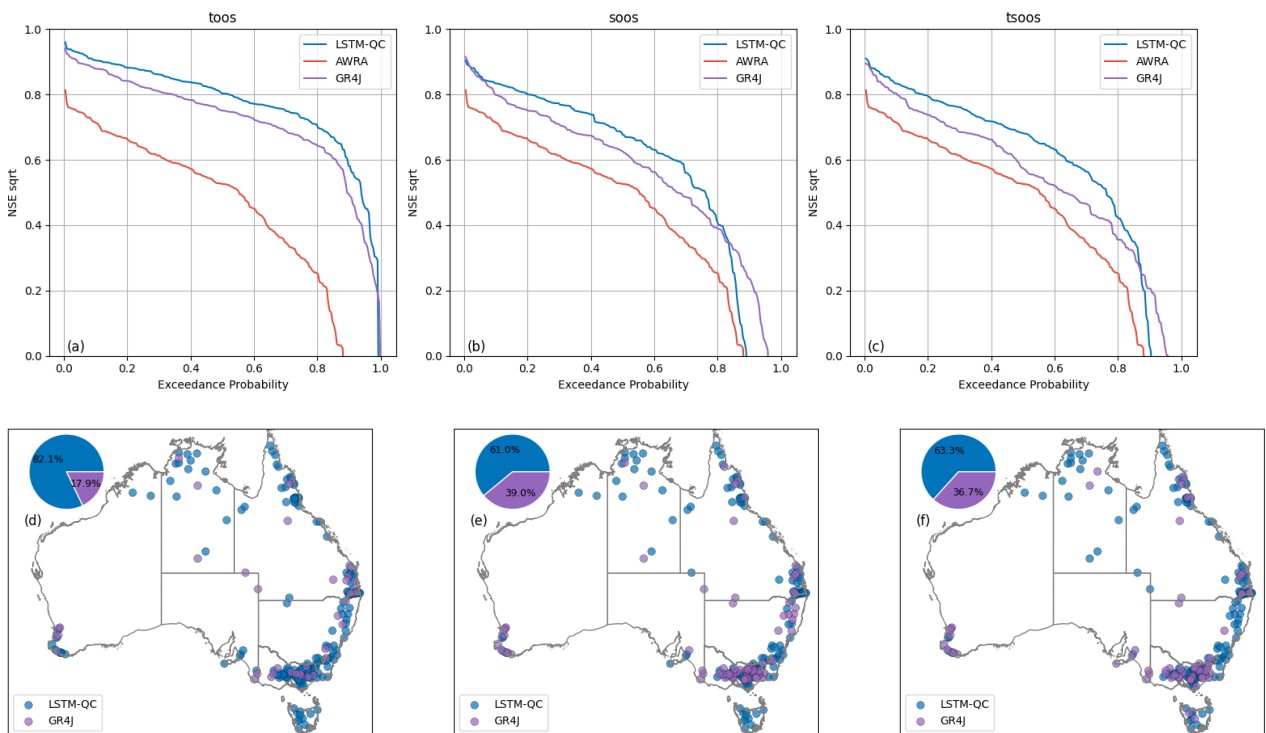

**Figure A1. Benchmarking LSTM-QC results against AWRA and GR4J. Top row is excedance curve of NSE sqrt across all catchments; bottom row shows which model performs best for each catchment. The columns from left to right show TooS, SooS, and TSooS cross-validation experiments**

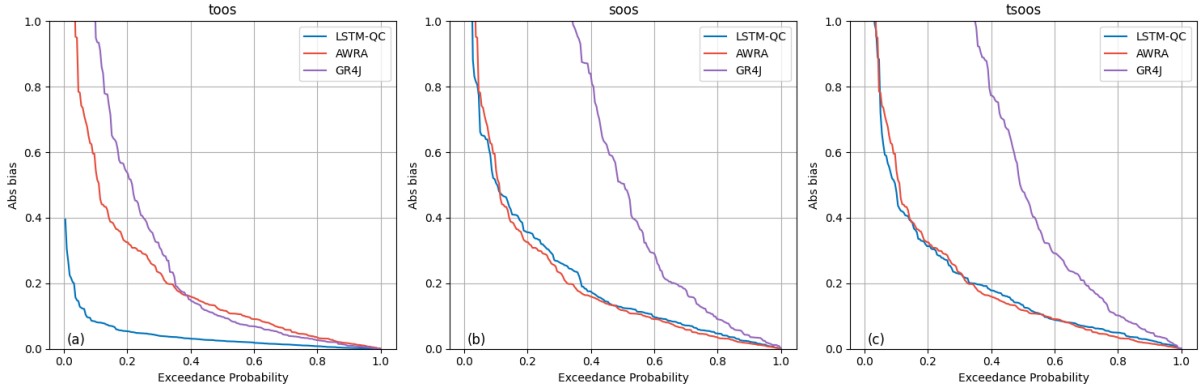

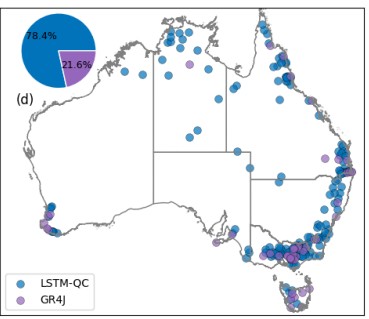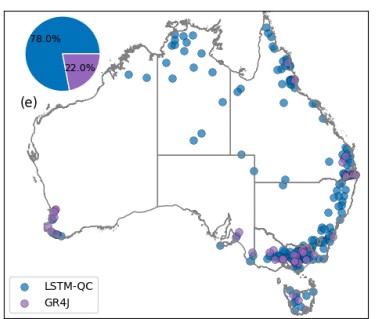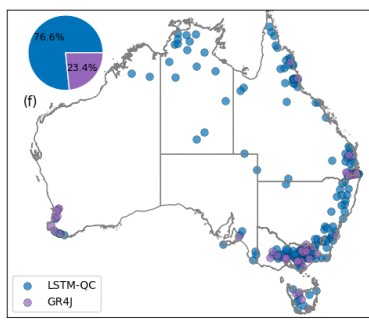

**Figure A2. Benchmarking LSTM-QC results against AWRA and GR4J. Top row is excedance curve of Absolute bias across all catchments; bottom row shows which model performs best for each catchment. The columns from left to right show TooS, SooS, and TSooS cross-validation experiments**

**Uncertainty Analysis:**

To assess the robustness of the reported results, we conducted two complementary analyses.

**Block bootstrap analysis:**

Hydrological time series exhibit strong temporal dependence, which can bias conventional resampling approaches. To address this, we implemented a block bootstrap with a block length of 365 days. For each model, 1000 bootstrap replicates were generated and performance metrics recalculated. The resulting confidence intervals provide an indication of sampling uncertainty while preserving the temporal structure of hydrological processes. This analysis confirms that the reported improvements are not an artefact of isolated hydrological events.

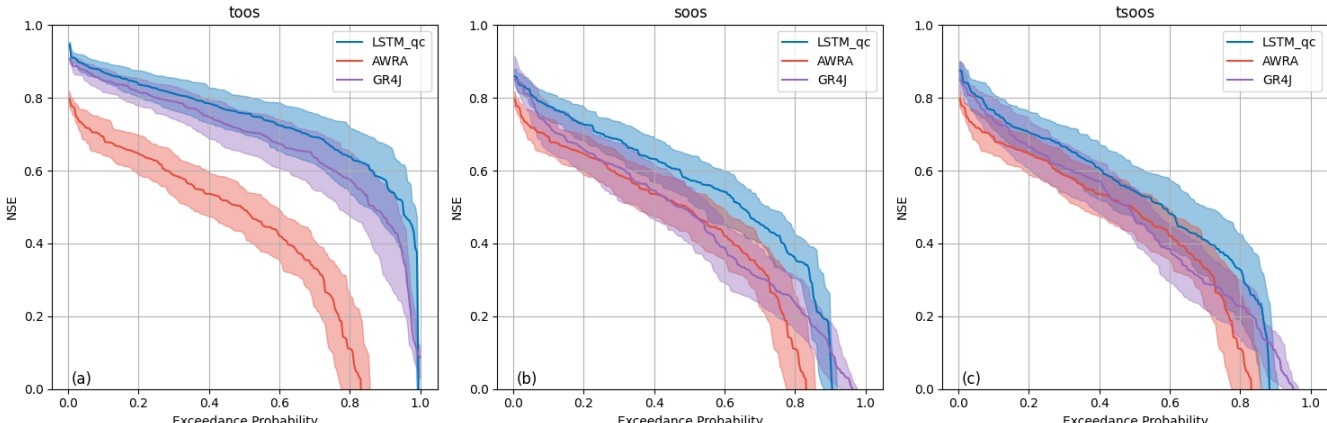

**Figure A3. Exceedance probability curves of NSE TooS, SooS, and TSooS experiments. Shaded regions show 95% confidence intervals from 1000 block bootstrap replicates (365-day blocks), confirming the robustness of LSTM-QC performance relative to AWRA-L and GR4J.**

**Sensitivity to model initialization and randomness:**

Deep learning models are inherently stochastic, with variability introduced by random initialization and optimization. To quantify this effect, we repeated each LSTM experiment 10 times using different random seeds. Median performance across runs is reported as the central curve, while the lighter blue lines represent the variability across trials. Although some spread is evident, the relative ordering of LSTM, AWRA-L, and GR4J is preserved across all repetitions.

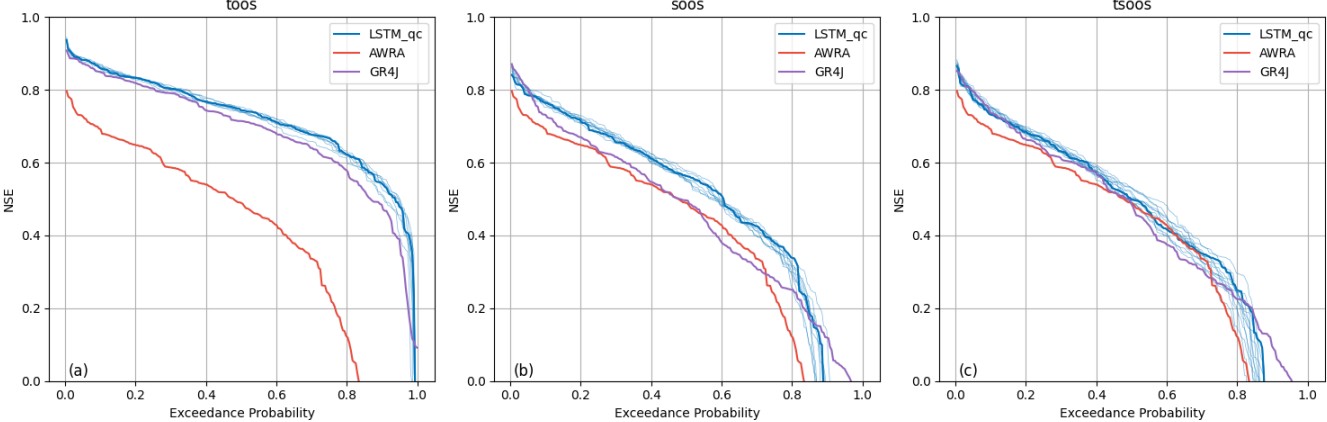

**Figure A4. NSE exceedance curves from TooS, SooS, and TSooS experiments. Thick blue line shows performance of median timeseries, and lighter blue lines show results from 10 independent LSTM runs with different random seeds.**