# Peer review of "Better continental-scale streamflow predictions for Australia: LSTM as a land surface model post-processor and standalone hydrological model"

_EGUsphere, 2025_

## Referee Comment (RC2)

**Review of *Better continental-scale streamflow predictions for Australia: LSTM as a land surface model post-processor and standalone hydrological model* by Shokri et al.**

June 5, 2025

**Overview**

This manuscript compares a standalone LSTM (LSTM-C) and an LSTM that includes simulated streamflow from a land surface model (AWRA-L) as an additional dynamic input (LSTM-QC). The two LSTM-based models are additionally benchmarked against AWRA-L and GR4J, a conceptual hydrological model widely used in an Australian context. They compare the models using three different cross-validation strategies, evaluating the ability of the models to predict temporally out-of-sample, spatially out-of-sample, and spatiotemporally out-of-sample. Overall, they show that the two LSTM models outperform AWRA-L and GR4J in most catchments under all three cross-validation strategies, although there are some notable exceptions. The authors discuss the potential relevance of their study in three real-world applications, namely historical reconstruction, predictions in ungauged basins, and simulating hydrological change under climate change projections.

**Main comments**

1. As far as I can tell, LSTM-C and LSTM-QC are identical in every respect except for the inclusion of streamflow from AWRA-L as an additional dynamic input in LSTM-QC. I therefore question whether this paper is really testing whether the LSTM can correct the AWRA-L output, or whether the AWRA-L output provides any additional information content that can be leveraged by the LSTM architecture. In effect these two possibilities amount to the same thing, but the paper would benefit from emphasising one over the other. In my opinion, the latter characterisation more accurately reflects what the LSTM is actually doing.

2. The results of the comparison with LSTM-C (i.e. the LSTM model that does not include the AWRA-L streamflow as a dynamic input) is undermined by the limited number of static catchment attributes that are supplied to the model (Table 1). A large number of studies have shown that LSTM models perform best when they are trained across many catchments at once using catchment descriptors that adequately describe the physical diversity of catchments in the training set. For example, (Kratzert et al., 2019) train an LSTM on static catchment attribtues that include soils, climate, vegetation, topography, and geology. Here, the authors have selected attributes that broadly cover climate and geomorphology, but discard a large number of attributes from CAMELS-AUS that are potentially highly influential in determining the hydrological behaviour of Australian catchments (e.g. geology, land cover). Presumably, in common with most land surface models, AWRA-L is parameterised using land cover and geological data. Therefore, I believe it is at least a possibility that LSTM-QC is utilising the information on catchment diversity that is encoded in the AWRA-L output but which has been arbitrarily excluded from LSTM-C. It seem to me that LSTM-C is trained in a way that is inconsistent with our current understanding of how best to use this class of model for hydrological simulation, raising doubts about whether it is a fair comparison.

**Minor comments**

1. L27: "The ubiquity of these model predictions..." - are you referring to the spatial coverage or widespread use? Please clarify.

2. L33: It's worth pointing out that most land surface models were not originally designed to predict streamflow, but rather to provide the lower boundary condition to Earth system models.

3. L39-48: I agree that the lack of channel routing and calibration scheme are weaknesses of AWRA-L with respect to streamflow simulation, but is a lack of process understanding not also a weakness?

4. L62: Punctuation needed.

5. L73: A third advantage is that they are unconstrained by physical laws such as mass balance, so they are better able to implicitly correct biases in the input data. In land surface models, uncertainty in the input will propagate to the output.

6. L85-93: This passage is not particularly relevant to the topic in hand. As the introduction is already quite long it could be safely removed.

7. L98: "...as we show in the current study..." - This would seem to pre-empt the results.

8. L99-100: Arguably deficiencies in routing and bias in individual catchments amount to the same thing. Perhaps you could clarify what you mean here?

9. L122: I'm not sure it is particularly easy to test the ability of the model to perform well under climate change projections, because it is likely that the range of input values in the climate projections will exceed those the LSTM would encounter in the training set. Thus what you really ought to be testing is the ability of the model to extrapolate, but I'm not sure the experimental design achieves this at present.

10. L156: You could acknowledge here that using multiple precipitation datasets can enhance LSTM performance (Kratzert et al., 2021).

11. L175: Please clarify that you are referring to the hidden state size here.

12. L221: You describe the static and dynamic predictors, but not the target (i.e. streamflow). Please could you describe your treatment of the target variable (e.g. do you normalize by catchment area)?

13. L225: Please could you confirm that the two LSTM models are identical in every respect except for the inclusion of AWRA-L streamflow in LSTM-QC?

14. L240: You say this important but not that you actually do it. We later find out that you have, although this information is in the results section. Please consider moving 3.1.3 to 2.5.2.

15. L245: In general I think the training approach for LSTMs is well established and so you don't need to go into so much detail here. The text could also be shortened by using scientific notation (e.g. Section 2.3 of (Lees et al., 2021)). Typically when training an LSTM there will be a training period, a validation period (that is used during training to test each parameter set) and a hold-out test period. However, Table 2 only details a training and validation period. Please could you clarify whether the model is tested on an unseen dataset?

16. L265: This needs some clarification. I think it is feasible (i.e. it could be done under the experimental setup) but not meaningful, because in a real out-of-sample situation you would not have any data to conduct fine-tuning.

17. L285: I can see the argument for including GR4J in the model comparison, but I wonder whether it would be better to only use it in the TooS test. I would argue that by including GR4J in the SooS and TSooS tests you are really testing the parameter regionalization scheme, which is not really the main focus of the manuscript.

18. Figure 4/5/6: Your description of the results would benefit from using subplot labels, so the reader knows what they should be looking at.

19. L373: Notwithstanding my previous point about climate projections, I'm not sure why this is categorised as TSooS rather than TooS?

20. L390: I'm not sure it is meaningful to compare with LSTM-C at short sequence lengths, as we already know that LSTMs require long sequence lengths to make good predictions.

21. L481: This could arise because the LSTM training is suboptimal, as it has not been exposed to catchment attributes that may help it learn the hydrological behaviour in these regions.

**References**

Kratzert, F., Klotz, D., Herrnegger, M., Sampson, A. K., Hochreiter, S., & Nearing, G. S. (2019). Toward Improved Predictions in Ungauged Basins: Exploiting the Power of Machine Learning. *Water Resour. Res.*, *55*(12), 11344–11354. https://doi.org/10.1029/2019WR026065

Kratzert, F., Klotz, D., Hochreiter, S., & Nearing, G. S. (2021). A note on leveraging synergy in multiple meteorological data sets with deep learning for rainfall–runoff modeling. *Hydrol. Earth Syst. Sci.*, *25*(5), 2685–2703. https://doi.org/10.5194/hess-25-2685-2021

Lees, T., Buechel, M., Anderson, B., Slater, L., Reece, S., Coxon, G., & Dadson, S. J. (2021). Benchmarking data-driven rainfall–runoff models in Great Britain: A comparison of long short-term memory (LSTM)-based models with four lumped conceptual models. *Hydrol. Earth Syst. Sci.*, *25*(10), 5517–5534. https://doi.org/10.5194/hess-25-5517-2021

---

## Author Response (AR1)

**Reviewer #1**

We thank the referee for their valuable comments. We have addressed each point in detail below and will incorporate the following changes:

**Summary**

**Comment:** *In this paper, the authors evaluate the performance of various LSTM models for hydrological simulations across Australian catchments: a land surface model-LSTM hybrid based on climate data as well as runoff from the AWRA-L land surface model, and an LSTM model based on climate data. They show that both models outperform runoff simulation from AWRA-L as well as from the conceptual hydrological model GR4J across most catchments. They investigate the impacts of methodological decisions, namely the cross-validation strategies, on the results. They additionally discuss the relevance of the proposed approches for three real-world applications: long-term historical simulations, predictions in ungauged basins, and climate projections. This application-focused framing is a welcomed perspective in a scientific paper. Overall, this is a well-executed, well-written study that addresses important research questions and contributes valuable insights to the hydrological modelling community. Below are some comments that will hopefully help review your paper for publication.*

**Response:** We sincerely appreciate your thorough and constructive review of our manuscript. Your positive assessment and insightful comments will help us improve both the clarity and depth of the study. In response, we have revised the manuscript to better articulate model design choices, clarified terminology, provided more context on catchment diversity, and addressed uncertainty and model evaluation procedures. Below, we address your comments point by point.

**Main Comments**

**Comment:** *The term "static features" or "static predictors" is somewhat misleading, especially since some of these predictors are recalculated for different time windows to demonstrate the impact on model performance. Please consider using a different term for the recalculated features (e.g., quasi-static) to remove any confusion.*

**Response:** We thank the reviewer for this helpful suggestion. We agree that the term "static predictors" may be misleading when referring to climatic attributes that are recalculated for different time windows in the temporal cross-validation experiments. To avoid confusion, we now distinguish between *static predictors* (geomorphological attributes that remain constant over time) and *quasi-static predictors* (climatic attributes such as mean annual precipitation and potential evapotranspiration, which are recalculated for each fold). This change has been implemented throughout the manuscript, including in Section 2.5.2, Table 1, Figure 5, and the Conclusion, to ensure consistent terminology.

**Comment:** *Model design clarity: It is unclear whether the static predictors are used for both LSTM-C and LSTM-QC models presented in section 2.5.1 by default, or only for some versions of these models. You could clarify this by merging sections 2.5.1 and 2.5.2 into a single model design section, summarizing model inputs more clearly (e.g., a table with two columns to*

*separate predictors into dynamic and static for each model). If static predictors are not always used and I misunderstood this, consider renaming the various versions of each model to differentiate them clearly.*

**Response:** We thank the reviewer for this helpful suggestion. To clarify, all static (geomorphological) and quasi-static (climatic) predictors are used in both LSTM-C and LSTM-QC models. We have revised the manuscript to make this explicit. As suggested, Sections 2.5.1 and 2.5.2 have been merged into a single "2.5.1 Model Design" section for improved clarity. We also added a new summary table (Table 3) that clearly distinguishes the dynamic, static, and quasi-static predictors used in each model configuration, with detailed definitions of the static and quasi-static predictors retained in Table 1.

**Comment:** *Catchment characteristics: It would be useful to know what the diversity of catchments is, for example using characteristics found in CAMELS-AU. This would help contextualize model transferability results (especially with SooS and TSooS) for an audience that is not familiar with Australian geography/hydrology.*

**Response:** We agree this would be helpful, especially for readers unfamiliar with Australian catchment diversity. We have now included a new figure (figure 1) showing the spatial distribution of CAMELS-AUS catchments overlaid with the Köppen–Geiger major climate zones (Stern et al, 2000).
Stern, H., De Hoedt, G., and Ernst, J.: Objective classification of Australian climates, Australian Meteorological Magazine, 49, 2000. https://www.weather-climate.com/AustMetMag2000pp87to96.pdf

**Comment:** *Uncertainty quantification: While comprehensive uncertainty analysis may be beyond the paper's scope, some quantification would enhance the robustness of the results, especially for hydroclimate projections or more specific applications such as regional-scale climate change vulnerability assessments (mentioned on L485-491). You could give an appreciation of the uncertainty for example by showing the spread in results from the cross-validation experiments (as mentioned on L267-268) in the Appendix. Additionally, you could apply bootstrapping when evaluating the model performance.*

**Response:** We thank the reviewer for highlighting the importance of assessing uncertainty and robustness in our modelling results. In response, we have added two complementary analyses in the Appendices: (i) a block bootstrap approach (365-day blocks) to account for temporal dependence and quantify uncertainty in performance metrics, and (ii) a sensitivity analysis across multiple random training trials to evaluate the variability due to model initialisation. The results (Figures A3–A4) show that the LSTM consistently outperforms AWRA-L and GR4J across resamples and trials.

**Comment:** *Model evaluation methodology: Please consider adding a dedicated subsection on model evaluation in the methods. This could include: i) an explanation of the NSE and what it measures (i.e., a measure of overall performance rather than an evaluation of extremes), and ii) the criterion for "best performing model" selection - was an ≥0.01 NSE difference sufficient or was there a more stringent measure (e.g., with a larger buffer)? My fear is that the results might be a bit noisy, and that using a more stringent measure or adding a statistical test to assess differences would be beneficial.*

**Response:** We thank the reviewer for this constructive suggestion. In the revised manuscript, we added a dedicated subsection on model evaluation in the Methods section, subsection 2.4.2, where we explain the NSE metric.

Regarding the criterion for selecting the "best performing model," we did not apply an arbitrary buffer but rather defined the best model at each catchment as the one with the highest NSE, we make it clear in the new Evaluation Metrics section. To address the reviewer's concern about noise and the magnitude of improvements, we emphasize that the top-row exceedance plots in Figure 4 provide a direct view of the distribution of NSE improvements across all catchments. These plots explicitly show the proportion of catchments that experience improvements greater than any given threshold. In this way, the results convey not only whether one model performs better than another, but also the scale and prevalence of those improvements across the full set of catchments. We clarify this by adding an explanation to Figure 4 result discription: *"In each catchment, the best-performing model is defined as the one with the highest NSE value. To avoid reliance on marginal differences, the exceedance curves also show the proportion of catchments where performance gains exceed any given threshold, providing a clearer picture of whether improvements are both consistent and substantial"*

**Comment:** *Terminology around climate projections: The term "climate projection" capabilities is somewhat misleading, as actual climate projections were not used here. Please consider reframing as "proxy for climate projection capabilities".*

**Response:** We agree with the reviewer that the original phrasing was misleading. We have revised the manuscript to consistently describe our evaluation as a "proxy for climate projection capabilities" rather than suggesting that actual climate projections were applied. This terminology has been updated in the Abstract, at the end of the Introduction, in the Results (first paragraph and Section 3.2.3 Application 3), and in the Conclusion.

**Specific comments:**

**Comment:** *L96-98: The "worthwhile" type of modelling system likely depends on the use. For example, for climate change scenarios the hybrid method might be favoured. Please clarify this nuance in the paper.*

**Response:** Thank you for pointing this out. We clarified this in the revised manuscript by adding the following sentences:

"The value of hybrid approaches may depend on the application context. For example, hybrid models may be particularly valuable for climate change scenario analysis, where maintaining physical consistency with land surface model outputs is important. Conversely, standalone LSTM models may be more advantageous for applications such as prediction in ungauged basins, where maximizing data-driven performance is the priority."

**Comment:** *L100-102: How can hybrid models help assess the dominant deficiencies? This warrant one to two more sentences in the paper.*

**Response:** Thank you for the suggestion. We revised the text to clarify how hybrid models can help assess dominant deficiencies in land surface models. Specifically, we explained that by comparing performance across different input sequence lengths, one can distinguish between improvements due to routing correction and those due to bias

correction. This diagnostic insight can assist land surface model developers in targeting specific weaknesses. The revised paragraph now reads:

*"In addition, in cases where LSTMs improve predictions from land surface models, as we show in the current study, the source of these improvements can be diagnosed. For instance, land surface models often exhibit two main deficiencies: routing errors and systematic biases in specific catchments. By comparing hybrid models trained with short input sequences (i.e. one time step) to those trained with longer sequences, we can isolate the contribution of each deficiency. Short sequence lengths limit the LSTM's capacity to correct routing errors, meaning improvements in this case are more likely due to bias correction."*

**Comment:** *L125: Is there any specific study of the application of GR4J in Australia that you could cite here?*

**Response:** Thank you for the suggestion. We added the following references to support the use of GR4J in the Australian context. Coron et al. (2012) provide a comprehensive evaluation of GR4J performance across 216 Australian catchments under diverse climate conditions. Hapuarachchi et al. (2022) describe the use of GR4J as part of the operational ensemble streamflow forecasting system for Australia. Zheng et al. (2024) further demonstrate the application of GR4J in projecting future streamflow under various climate change scenarios for Australia.
- Coron, L., Andréassian, V., Perrin, C., Lerat, J., Vaze, J., Bourqui, M., & Hendrickx, F. (2012). Crash testing hydrological models in contrasted climate conditions: An experiment on 216 Australian catchments. Water Resources Research, 48, W05552.
- Hapuarachchi, H. A. P., Bari, M. A., Kabir, A., Hasan, M. M., Woldemeskel, F. M., Gamage, N., Sunter, P. D., Zhang, X. S., Robertson, D. E., Bennett, J. C., & Feikema, P. M. (2022). Development of a national 7-day ensemble streamflow forecasting service for Australia. Hydrology and Earth System Sciences, 26, 4801–4821.
- Zheng, H., Chiew, F. H. S., Post, D. A., Robertson, D. E., Charles, S. P., Grose, M. R., & Potter, N. J. (2024). Projections of future streamflow for Australia informed by CMIP6 and previous generations of global climate models. *Journal of Hydrology*, 636, 131286.

**Comment:** *L225: Please explicitly mention what the AWRA-L output is here to remind the reader.*

**Response:** Thank you for this recommendation. We have revised the text to clearly state that the AWRA-L output used in our study refers to gridded runoff (surface and subsurface) at a 5 km × 5 km resolution across Australia. Specifically, we added the clarification *"(i.e., gridded runoff from surface and subsurface processes at a 5 km × 5 km resolution across Australia)"* to the Model design section (2.5.1).

**Comment:** *L280-283: The differences seen using an increasing time window could also be impacted by the catchment size, with larger differences between the two performance measures expected in larger catchments. It would be interesting to compare the length of the sequence with the known response time of each catchment.*

**Response:** Thank you for this suggestion. We examined how catchment size influences the performance gains from including AWRA-L runoff (Qtot), measured as the difference between ($NSE_{LSTM-QC} - NSE_{LSTM-C}$), across different sequence lengths (1 day vs. 365

days). Our analysis shows that AWRA-L inclusion yields larger improvements in NSE for bigger catchments when a short sequence length (1 day) is used. This effect arises because AWRA-L incorporates a partial surface water storage at the grid-cell level. While this does not resolve in-stream routing, it does simulate some delayed runoff processes within each pixel. When used as input to the LSTM, this storage effect provides additional memory that helps improve predictions for large catchments. However, as the sequence length provided to the LSTM increases, the model itself is able to capture these dependencies, and the added benefit of AWRA-L diminishes.

[Figure]

Given that this was primarily an illustrative exercise (since we would not recommend using a 1-day sequence length in practice for models like LSTM-C), we have chosen not to include this additional figure in the manuscript to avoid overcomplicating the presentation. Instead, we now note in the Discussion "… We also note that the influence of sequence length is partly modulated by catchment size: larger catchments tend to benefit more from the inclusion of AWRA-L runoff at very short sequence lengths, as its partial surface water storage provides additional memory. However, this advantage diminishes as longer sequences allow the LSTM itself to capture the relevant dependencies".

**Comment:** *L312: "higher exceedance probabilities" might be misinterpreted as referring to flow exceedance. Please clarify that this refers to the distribution of values when introducing the first plot of this kind.*

**Response:** We clarified that this refers to the distribution of NSE values across catchments, not to flow exceedance:
"…the fine-tuned model (blue) consistently outperformed the global model (red), especially at higher exceedance probabilities of NSE values across catchments."

**Comment:** *L312-314: It would be interesting to speculate why some catchments don't benefit from finetuning. Are there commonalities in catchment type, data quality, hydro-meteorological processes, etc?*

**Response:** Thanks for the suggestion. In our analysis, only four catchments showed reduced performance after fine-tuning. These catchments are either ephemeral rivers with highly variable and intermittent flow (meaning they can cease to flow for many

consecutive years and experience very large, infrequent flow events when they do occur) or are characterised by limited data. Such conditions make them difficult to model, and due to their variability, local fine-tuning may overfit or underperform relative to a more generalized global model. We will expand on this point in the revised manuscript, providing additional detail on the characteristics of these catchments. This explanation is added to section 3.1.1.

**Comment:** *L388-389: Please consider spelling out what you mean by "possibly other hydrological processes" in the paper. For example, groundwater storage and processes, as well as lakes and reservoirs could be mentioned here. A side question to this, are the effects of lakes and reservoirs accounted for by the AWRA model?*

**Response:** We acknowledge the need for greater clarity here. In the revised manuscript, we specified the types of hydrological processes we are referring to: "*…showing an improvement in performance metrics, which is mostly due to channel routing processes and additional lag processes such as percolation, groundwater interactions, and human influences (e.g., farm dams)*".

Regarding the side question: the AWRA model does not explicitly simulate lakes or large reservoirs. However, the catchments used in this study are not impacted by major reservoirs. Nonetheless, small farm dams are present and may impact local hydrology, particularly by modifying runoff and storage patterns. While not directly modelled, their effects are likely implicitly represented through calibration where data are available. These farm dams have a widespread and growing impact on water availability across Australian agricultural regions and can significantly reduce downstream flows, particularly during dry years. We explained this in the revised manuscript introduction as follows:

"AWRA-L also does not explicitly simulate lakes or large reservoirs. While the catchments used in this study are not impounded by major reservoirs and were nominally selected to avoid the impact of farm dams (see Zhang et al, 2013), small farm dams are widespread across many agricultural regions of Australia. These can significantly alter runoff and storage patterns, particularly during dry years, by reducing downstream flows. Although farm dams are not directly represented in AWRA-L, their effects are likely partially captured through calibration where observational data are available and farm dams were present . Recent studies have highlighted the growing regional impact of farm dams on water availability under climate change (Malerba et al., 2021; Peña-Arancibia et al., 2023).

- Peña-Arancibia, J. L., M. E. Malerba, N. Wright and D. E. Robertson (2023). "Characterising the regional growth of on-farm storages and their implications for water resources under a changing climate." Journal of Hydrology 625: 130097.
- Malerba, M. E., N. Wright and P. I. Macreadie (2021). "A Continental-Scale Assessment of Density, Size, Distribution and Historical Trends of Farm Dams Using Deep Learning Convolutional Neural Networks." Remote Sensing 13(2): 319.
- Zhang, Y., N. Viney, A. Frost, A. Oke, M. Brooks, Y. Chen and N. Campbell (2013). "Collation of Australian modeller's streamflow dataset for 780 unregulated Australian catchments." CSIRO, Australia. csiro:EP113194.
https://doi.org/10.4225/08/58b5baad4fcc2

**Comment:** *L492: One advantage of land surface or conceptual hydrological models compared to LSTM models is that they can output various other hydrological variables in addition to streamflow. Please considering adding this to the list.*

**Response:** We agree and added that conceptual and land surface models can output additional hydrological variables to the last paragraph of the conclusion.

**Typos:**

*L62: Some kind of sentence separation is needed between "streamflow" and "for instance".*

*L114: Missing "in" or similar between "performance" and "218 catchments".*

*L133: In the introduction it says that outputs are available from 1910. -> introduction is fixed*

*L138: Missing closing parenthesis.*

*L152: Missing "for catchments" or similar between ")" and "that have been".*

*L154: Rephrase "covering from".*

*L157: "is produce" is missing a "d".*

*L263: Missing "the" between "from" and "calibration".*

*L469: Small "b" for "behavior".*

*L472: "strongly performing"?*

**Response:** Thank you for identifying these issues. We corrected all of them in the revised manuscript.

**Reviewer #2**

We appreciate the reviewer's careful reading and constructive feedback. We address each comment below and describe the revisions we will incorporate into the manuscript.

**Overview:**

***Comment:*** *This manuscript compares a standalone LSTM (LSTM-C) and an LSTM that includes simulated streamflow from a land surface model (AWRA-L) as an additional dynamic input (LSTM-QC). The two LSTM-based models are additionally benchmarked against AWRA-L and GR4J, a conceptual hydrological model widely used in an Australian context. They compare the models using three different cross-validation strategies, evaluating the ability of the models to predict temporally out of-sample, spatially out-of-sample, and spatiotemporally out-of-sample. Overall, they show that the two LSTM models outperform AWRA-L and GR4J in most catchments under all three cross-validation strategies, although there are some notable exceptions. The authors discuss the potential relevance of their study in three real-world applications, namely historical reconstruction, predictions in ungauged basins, and simulating hydrological change under climate change projections.*

> **Response**: We thank you for the clear and thoughtful summary of our manuscript. Your overview accurately captures the key elements of our study and reflects our intent to assess the relative performance of different modelling approaches under varying out-of-sample conditions. We hope the revisions we have made in response to your detailed comments further clarify the study's motivations, methodology, and implications.

**Main comments**

***Comment:*** *1. As far as I can tell, LSTM-C and LSTM-QC are identical in every respect except for the inclusion of streamflow from AWRA-L as an additional dynamic input in LSTM-QC. I therefore question whether this paper is really testing whether the LSTM can correct the AWRA-L output, or whether the AWRA-L output provides any additional information content that can be leveraged by the LSTM architecture. In effect these two possibilities amount to the same thing, but the paper would benefit from emphasising one over the other. In my opinion, the latter characterisation more accurately reflects what the LSTM is actually doing.*

> **Response:** We agree with the reviewer that the distinction between correcting AWRA-L output versus leveraging its additional information content is important, and we appreciate this opportunity to clarify the framing. In the revised manuscript, we:
> (1) Re-casted the Introduction and Abstract to emphasise that our primary goal is to assess the information content of AWRA-L when used as a predictor, rather than to correct its outputs. We now explicitly state that LSTM-QC and LSTM-C differ only by the inclusion of AWRA-L runoff, and that our comparison quantifies the added value of this information.
> **Revised text:**
> *"LSTM-QC, a rainfall–runoff LSTM that incorporates runoff outputs from the Australian Water Resources Assessment–Landscape model (AWRA-L), which can also be interpreted as a post-processor"*
> *"interpreted as the* information content *provided by the land surface model. Land surface models often exhibit two main deficiencies: routing errors and systematic biases in specific catchments. By comparing hybrid models trained with short input sequences (i.e. one time step) to those trained with longer sequences, we can distinguish the type of information*

*AWRA-L contributes. Short sequence lengths limit the LSTM's ability to correct routing errors, meaning improvements in this case are more likely due to information related to bias correction."*

(2) Updated the Discussion by re-labelling Figures 4 and 7 and revising the text to make sure it is reflecting this fact better.

**Revised opening paragraph of the Discussion:** "The results of this study highlight the significant potential of LSTM networks in streamflow prediction as a rainfall-runoff model (LSTM-C) or as a rainfall-runoff model that incorporates additional information from the AWRA-L land surface model (LSTM-QC), which can also be viewed as a post-processor for AWRA-L."

**Revised sequence length description:** "The sensitivity of the LSTM-QC and LSTM-C to the length of predictors passed to the model was investigated, enabling a decomposition of the information provided by AWRA-L in terms of bias-related signals and routing-related signals."

(3) Revised the Conclusions to highlight that the main contribution of this work lies not in correcting AWRA-L biases, but in evaluating the informational value of a process-based model (AWRA-L) when used alongside data-driven LSTM approaches.

**Revised first paragraph of the Conclusions:** "The potential of Long Short-Term Memory (LSTM) networks to enhance streamflow predictions in Australia was evaluated. The findings demonstrate that LSTM networks—whether functioning as standalone rainfall–runoff models (LSTM-C) or as rainfall–runoff models that incorporate additional information from AWRA-L (LSTM-QC) and thus can also be interpreted as post-processors to AWRA-L—consistently improve prediction accuracy across Australia relative to existing models."

*Comment: 2. The results of the comparison with LSTM-C (i.e. the LSTM model that does not include the AWRA-L streamflow as a dynamic input) is undermined by the limited number of static catchment attributes that are supplied to the model (Table 1). A large number of studies have shown that LSTM models perform best when they are trained across many catchments at once using catchment descriptors that adequately describe the physical diversity of catchments in the training set. For example, (Kratzert et al., 2019) train an LSTM on static catchment attributes that include soils, climate, vegetation, topography, and geology. Here, the authors have selected attributes that broadly cover climate and geomorphology, but discard a large number of attributes from CAMELS-AUS that are potentially highly influential in determining the hydrological behaviour of Australian catchments (e.g. geology, land cover). Presumably, in common with most land surface models, AWRA-L is parameterised using land cover and geological data. Therefore, I believe it is at least a possibility that LSTM-QC is utilising the information on catchment diversity that is encoded in the AWRA-L output but which has been arbitrarily excluded from LSTM-C. It seem to me that LSTM-C is trained in a way that is inconsistent with our current understanding of how best to use this class of model for hydrological simulation, raising doubts about whether it is a fair comparison.*

**Response:** Thank you for your thoughtful comment. We agree that the choice of static catchment attributes can significantly affect LSTM model performance. The selection of static features in our study was deliberate and guided by both performance considerations and data availability in operational or ungauged settings.

As noted in Kratzert et al. (2019), they also did not use the full set of static attributes

available in CAMELS. Instead, they selected 27 features as a subset of those explored by Addor et al. (2017), focusing on variables derivable from remote sensing or nationally available datasets. Similarly, in our case, we explored a wide range of attributes available in CAMELS-AUS and found through systematic testing that a subset of 12 variables consistently improved model performance across catchments. These variables cover key aspects of climate and geomorphology and were chosen to ensure applicability in real-world, data-limited contexts.

We also deliberately excluded static variables derived from streamflow to avoid highly correlated predictors, and we omitted attributes that are difficult to estimate reliably for ungauged basins. This aligns with our focus on creating a parsimonious and operationally feasible model.

**Minor comments**

*Comment: 1. L27: "The ubiquity of these model predictions..."- are you referring to the spatial coverage or widespread use? Please clarify.*

**Response:** Thank you for pointing this out. In this context, "ubiquity" refers primarily to the widespread spatial coverage of land surface model predictions across large regions, often at continental or global scales. We revised the sentence to clarify this and remove ambiguity.
"The widespread spatial coverage of these model predictions often trades off against accuracy..."

*Comment: 2. L33: It's worth pointing out that most land surface models were not originally designed to predict streamflow, but rather to provide the lower boundary condition to Earth system models.*

**Response:** We agree that many land surface models were originally developed to provide lower boundary conditions for Earth system models rather than for direct streamflow prediction. However, we would like to clarify that unlike most other land surface models AWRA-L was specifically developed for water balance estimation and runoff prediction across Australia, with a focus on hydrological applications rather than atmospheric coupling. We clarified this distinction in the revised manuscript by adding:
*"It is worth noting that while many land surface models were originally developed to provide boundary conditions for Earth system models rather than for direct streamflow prediction, AWRA-L was specifically designed for water balance estimation and runoff prediction across Australia, with an emphasis on hydrological applications rather than atmospheric coupling, and calibrated to streamflow observations."*

*Comment: 3. L39-48: I agree that the lack of channel routing and calibration scheme are weaknesses of AWRA-L with respect to streamflow simulation, but is a lack of process understanding not also a weakness?*

**Response:** We agree that understanding and simulating key processes adds to confidence in a model, and that the sometimes-imperfect representation of these processes in AWRA-L (as well as the total lack of such processes in LSTMs), may contribute to a lack of confidence in these models. We updated the manuscript to explicitly acknowledge the sometimes-imperfect representation of hydrological processes of AWRA-L, alongside the issues of channel routing and calibration.

*"catchments. In addition, the sometimes-imperfect representation of hydrological processes in AWRA-L reduces confidence in its streamflow predictions. "*

**Comment:** *4. L62: Punctuation needed.*

**Response:** Thank you for pointing this out. We revised the sentence for clarity and correct punctuation.

**Comment:** *5. L73: A third advantage is that they are unconstrained by physical laws such as mass balance, so they are better able to implicitly correct biases in the input data. In land surface models, uncertainty in the input will propagate to the output.*

**Response:** Thank you for the suggestion. We agree and have incorporated this point into the revised manuscript. The paragraph has been updated as follows:
"… A third advantage is that they are not constrained by physical laws such as mass balance, which allows them to implicitly correct biases in the input data. In contrast, in land surface models, uncertainty in the inputs typically propagates directly to the outputs."

**Comment:** *6. L85-93: This passage is not particularly relevant to the topic in hand. As the introduction is already quite long it could be safely removed.*

**Response:** Thank you for the suggestion. We removed this part.

**Comment:** *7. L98: "...as we show in the current study..."- This would seem to pre-empt the results.*

**Response:** Thank you for pointing this out. We removed "as we show in the current study" to maintain a more neutral tone in the introductory text.

**Comment:** *8. L99-100: Arguably deficiencies in routing and bias in individual catchments amount to the same thing. Perhaps you could clarify what you mean here?*

**Response:** While both routing deficiencies and catchment-specific biases contribute to model error, they stem from different sources and have distinct implications for model behaviour. Routing deficiencies primarily affect the timing and shape of the hydrograph (e.g., delayed or premature peak flows), whereas biases typically refer to systematic over- or underestimation of flow magnitude, independent of timing. This distinction is important for diagnosing model limitations. We updated the manuscript to more clearly articulate this difference:
*"Routing errors primarily affect the timing and shape of the hydrograph, while systematic biases reflect consistent over- or underestimation of flow magnitude, regardless of timing."*

**Comment:** *9. L122: I'm not sure it is particularly easy to test the ability of the model to perform well under climate change projections, because it is likely that the range of input values in the climate projections will exceed those the LSTM would encounter in the training set. Thus, what you really ought to be testing is the ability of the model to extrapolate, but I'm not sure the experimental design achieves this at present.*

**Response:** Thank you for raising this important point. We agree that true testing under climate change conditions requires the model to extrapolate beyond the historical range

of climate inputs, which is inherently challenging. While our experimental setup does not fully replicate future climate scenarios, the Temporally and Spatially out of Sample (TSooS) cross-validation partially addresses this concern. In this design, the data are split into spatiotemporal quadrants, such that the model is trained on one period (e.g. 1975–1995) and evaluated on a different period (e.g. 2000–2014) in distinct catchments. This setup introduces a degree of extrapolation in both space and time. However, we acknowledge that the future climate may involve more extreme conditions than those seen in our historical training period. To better reflect this limitation, we have revised the manuscript to refer to this analysis as a "proxy for climate projection capabilities," rather than implying direct applicability to future climate conditions.

**Comment:** *10. L156: You could acknowledge here that using multiple precipitation datasets can enhance LSTM performance (Kratzert et al., 2021).*

**Response:** Thank you for the suggestion. While previous studies such as Kratzert et al. (2021) have shown that using multiple precipitation datasets can enhance LSTM performance, in our case the AGCD (formerly known as AWAP) and SILO datasets share many common rain gauges and differ mainly in processing methods. As a result, incorporating both datasets may not improve model performance greatly. However, the main reason for omitting SILO is to test our methods under close-to operational conditions. AGCD is produced operationally at the Bureau, and is available in real time. While many of the gauges used in AGCD are also used by SILO, AGCD uses fewer rain gauges than SILO, which incorporates many now-defunct rainfall gauges in its analysis. This means that if the LSTM performance did improve with SILO, these improvements would not have been available if the model was run in operations, as these rain gauges are no longer available. AGCD is much less sensitive to rain gauges becoming unavailable. See Fawcett R., Trewin B. and Barnes-Keoghan I. (2010) "Network-derived inhomogeneity in monthly rainfall analyses over western Tasmania", *17th National Conference of the Australian Meteorological and Oceanographic Society*. Canberrra. doi: https://doi.org/10.1088/1755-1315/11/1/012006

**Comment:** *11. L175: Please clarify that you are referring to the hidden state size here.*

**Response:** Thank you for the comment. We clarified in the text that the reference is specifically to the hidden state size.

**Comment:** *12. L221: You describe the static and dynamic predictors, but not the target (i.e. streamflow). Please could you describe your treatment of the target variable (e.g. do you normalize by catchment area)?*

**Response:** Good observation. The target is gauged daily streamflow observation which is provided in the CAMELS-AUS dataset and is normalized by catchment area in mm unit. We added this to the text.

**Comment:** *13. L225: Please could you confirm that the two LSTM models are identical in every respect except for the inclusion of AWRA-L streamflow in LSTM-QC?*

**Response:** Yes, we can confirm that the LSTM models are identical in every respect except for the use of AWRA-L runoff as a predictor in LSTM-QC. We have revised the model

design section and stated this explicitly in the text (Section 2.5.1 and Table 3 in revised manuscript).

**Comment:** *14. L240: You say this important but not that you actually do it. We later find out that you have, although this information is in the results section. Please consider moving 3.1.3 to 2.5.2.*

**Response:** Thank you for pointing this out. We have comprehensively revised Sections 2.5.1 and 2.5.2 of the original manuscript and merged them into a new section, "2.5.1 Model Design", with a clear explanation of the recalculation of static predictors. In addition, to avoid confusion, we now refer to predictors that require recalculation as quasi-static predictors.

**Comment:** *15. L245: In general I think the training approach for LSTMs is well established and so you don't need to go into so much detail here. The text could also be shortened by using scientific notation (e.g. Section 2.3 of (Lees et al., 2021)). Typically when training an LSTM there will be a training period, a validation period (that is used during training to test each parameter set) and a hold-out test period. However, Table 2 only details a training and validation period. Please could you clarify whether the model is tested on an unseen dataset?*

**Response:** We appreciate this useful suggestion. Regarding the first point: In the revised manuscript, we have shortened this section and adopted the scientific shorthand style recommended (e.g., Lees et al., 2021, Section 2.3).
Regarding the hold-out test period: You are correct that Table 2 currently lists only the training and validation periods. In fact, we also use a separate hold-out test period (2014–2022), with model performance illustrated by the dashed line in Figure 4. We added this explanation to the section 2.4.1.

**Comment:** *16. L265: This needs some clarification. I think it is feasible (i.e. it could be done under the experimental setup) but not meaningful, because in a real out-of-sample situation you would not have any data to conduct fine-tuning.*

**Response:** Thank you for the comment. We agree that the term "feasible" may be misleading in this context. While fine-tuning is technically possible within the experimental setup, it would not be meaningful in a true out-of-sample scenario where no data from the target catchment would be available for adjustment. We revised the manuscript accordingly and replace "feasible" with "realistic" to better reflect the intention. The revised sentence will read:
*"Fine-tuning for individual catchments would not be realistic in a true out-of-sample scenario, as no target catchment data would be available for adjustment."*

**Comment:** *17. L285: I can see the argument for including GR4J in the model comparison, but I wonder whether it would be better to only use it in the TooS test. I would argue that by including GR4J in the SooS and TSooS tests you are really testing the parameter regionalization scheme, which is not really the main focus of the manuscript.*

**Response:** While we understand the concern, we maintain that testing GR4J under the SooS and TSooS setups aligns with one of our primary objectives, which is to evaluate model performance in regionalization scenarios. Including GR4J across all experimental setups allows for a consistent benchmark against a widely used conceptual model, noting

that GR4J is also used in out-of-spatial-sample prediction in Australia. This helps illustrate the value of LSTM models in both interpolation and extrapolation contexts.

*Comment: 18. Figure 4/5/6: Your description of the results would benefit from using subplot labels, so the reader knows what they should be looking at.*

**Response:** We thank the reviewer for pointing this out. To improve clarity, we have revised the results text for Figures 4–6 so that it explicitly directs the reader to what they should be looking at in each subplot (e.g., trends in exceedance curves or spatial patterns).

*Comment: 19. L373: Notwithstanding my previous point about climate projections, I'm not sure why this is categorised as TSooS rather than TooS?*

**Response:** Thank you for your comment. The reason this is categorized as TSooS rather than TooS relates to the data partitioning strategy used for training and testing. In the TooS setup, the model is trained on all catchments but only during one period, then tested on the same catchments during a different period. In contrast, TSooS is a stricter test where the model is trained on only half of the catchments for half of the overall time period and then tested on the remaining unseen catchments during the other (also unseen) half of the time. This means the model effectively trains on only about a quarter of the total data in TSooS, compared to about three-quarters in TooS. This distinction is important because TSooS better evaluates the model's ability to generalize to completely new catchments and unseen periods, which may include extreme events like the millennium drought that are absent from the training data. To clarify this, we added following explanation to the text: *"This experiment is categorized as TSooS because validation data are independent in both space and time. Specifically, the model is trained on half of the catchments over half of the available period and tested on the other half of the catchments during the remaining period. This setup ensures that the validation set comprises entirely unseen catchments and time periods, providing a more stringent test of model generalization than TooS and SooS."*

*Comment: 20. L390: I'm not sure it is meaningful to compare with LSTM-C at short sequence lengths, as we already know that LSTMs require long sequence lengths to make good predictions.*

**Response:** We agree that it is already known that LSTM-C requires longer sequence lengths to perform well. However, we use LSTM-C here primarily as a baseline to demonstrate the added value of the routing component in LSTM-QC. Even at shorter sequence lengths, the difference in performance between LSTM-C and LSTM-QC highlights the extent to which AWRA-L is resolving routing processes. We believe this comparison offers important insights for AWRA-L users. To clarify this, the following explanation has been added to the analysis of Figure 7: *"The comparison with LSTM-C at very short sequence lengths is not intended as a practical configuration for standalone LSTMs, which are known to require longer sequences to perform well. Instead, it serves as a baseline to illustrate the added value of AWRA-L runoff: even when the sequence length is too short for LSTM-C to capture catchment memory, LSTM-QC can leverage AWRA-L to resolve routing processes."*

***Comment:*** *21. L481: This could arise because the LSTM training is suboptimal, as it has not been exposed to catchment attributes that may help it learn the hydrological behaviour in these regions.*

**Response**: We agree that suboptimal LSTM training due to limited exposure to catchment attributes is one possible explanation. However, we believe the poorer performance of LSTMs compared to GR4J in south-west Western Australia is more likely due to the more informative regionalisation scheme used by GR4J. This region is hydrologically distinct (Petrone et al., 2010; Hughes et al., 2012), and since GR4J's regionalisation is weighted by inverse-distance, it places greater emphasis on local catchments, whereas the LSTM does not. That said, it is possible that training the LSTM on a more global dataset might improve its performance in this region. We consider a thorough investigation of these hypotheses outside the scope of the current paper and intend to address them in future research. We added this discussion to the revised manuscript.

Petrone, K. C., J. D. Hughes, T. G. Van Niel, and R. P. Silberstein (2010), Streamflow decline in southwestern Australia, 1950–2008, Geophys. Res. Lett., 37, L11401, doi:10.1029/2010GL043102.

Hughes, J. D., K. C. Petrone, and R. P. Silberstein (2012), Drought, groundwater storage and stream flow decline in southwestern Australia, Geophys. Res. Lett., 39, L03408, doi:10.1029/2011GL050797.

---

## Author Response (AR3)

Dear Editor,

Thank you for your notification and for the final checks on our manuscript.
Regarding the editorial notifications made by Polina Shvedko:
- The two previously separate "Code and data availability" sections have now been merged into a single section under the headline "Code and data availability".
- The corresponding author is affiliated with the Commonwealth Scientific and Industrial Research Organisation (CSIRO). The ROR listing correctly identifies CSIRO, with Canberra listed as the main office, while the corresponding author is based at the Clayton campus.

Kind regards,
Ashkan Shokri